# Generalized Depthwise-Separable Convolutions for Adversarially Robust and Efficient Neural Networks

**Hassan Dbouk & Naresh R. Shanbhag**
Department of Electrical and Computer Engineering
University of Illinois at Urbana-Champaign
Urbana, IL 61801
{hdbouk2,shanbhag}@illinois.edu

## Abstract

Despite their tremendous successes, convolutional neural networks (CNNs) incur high computational/storage costs and are vulnerable to adversarial perturbations. Recent works on robust model compression address these challenges by combining model compression techniques with adversarial training. But these methods are unable to improve throughput (frames-per-second) on real-life hardware while simultaneously preserving robustness to adversarial perturbations. To overcome this problem, we propose the method of Generalized Depthwise-Separable (GDWS) convolution – an *efficient, universal, post-training* approximation of a standard 2D convolution. GDWS dramatically improves the throughput of a standard pre-trained network on real-life hardware while preserving its robustness. Lastly, GDWS is scalable to large problem sizes since it operates on pre-trained models and doesn't require any additional training. We establish the optimality of GDWS as a 2D convolution approximator and present exact algorithms for constructing optimal GDWS convolutions under complexity and error constraints. We demonstrate the effectiveness of GDWS via extensive experiments on CIFAR-10, SVHN, and ImageNet datasets. Our code can be found at https://github.com/hsndbk4/GDWS.

## 1 Introduction

Nearly a decade of research after the release of AlexNet [18] in 2012, convolutional neural networks (CNNs) have unequivocally established themselves as the *de facto* classification algorithm for various machine learning tasks [11, 38, 4]. The tremendous success of CNNs is often attributed to their unrivaled ability to extract correlations from large volumes of data, allowing them to surpass human level accuracy on some tasks such as image classification [11].

Today, the deployment of CNNs in safety-critical Edge applications is hindered due to their **high computational costs** [11, 30, 31] and their **vulnerability** to adversarial samples [37, 5, 16]. Traditionally, those two problems have been addressed in isolation. Recently, very few bodies of works [19, 35, 42, 6, 34, 7] have addressed the daunting task of designing both **efficient** and **robust** CNNs. A majority of these methods focus on model compression, i.e. reducing the storage requirements of CNNs. None have demonstrated their real-time benefits in hardware. For instance, Fig. 1a shows recent robust pruning works HYDRA [34] and ADMM [42] achieve high compression ratios (up to $97\times$) but either *fail* to achieve high throughput measured in frames-per-second (FPS) or *compromise* significantly on robustness. Furthermore, the overreliance of current robust complexity reduction techniques on adversarial training (AT) [45, 21] increases their training time significantly (Fig. 1b). This prohibits their application to complex ImageNet scale problems with stronger attack models, such as union of norm-bounded perturbations [22]. Thus, there is critical need for methods to

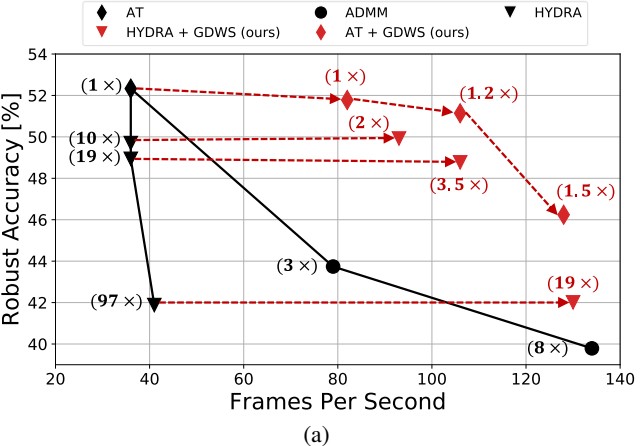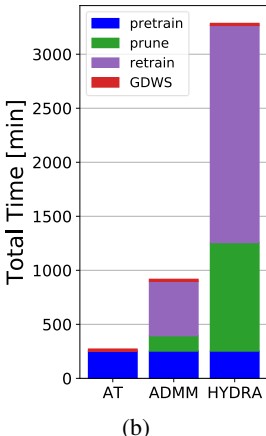

| (a) | (b) |

Figure 1: Performance of existing robust pruning works (HYDRA [34] and ADMM [42]) and the proposed GDWS with VGG-16 on CIFAR-10, captured by: (a) robust accuracy against $\ell_\infty$-bounded perturbations vs frames-per-second measured on an NVIDIA Jetson Xavier, and (b) total time required to implement these methods measured on a single NVIDIA 1080 Ti GPU. To ensure a fair comparison, the same AT baseline (obtained from [34]) is used for all methods. The compression ratio of each method, highlighted in parenthesis, is with respect to the AT baseline.

design deep nets that are both adversarially robust and achieve high throughput when mapped to real hardware.

To address this need, we propose **Generalized Depthwise-Separable (GDWS)** convolutions, a *universal post-training* approximation of a standard 2D convolution that dramatically improves the real hardware FPS of pre-trained networks (Fig. 1a) while preserving their robust accuracy. Interestingly, we find GDWS applied to un-pruned robust networks simultaneously achieves higher FPS and higher robustness than robust pruned models obtained from current methods. This in spite of GDWS's compression ratio being smaller than those obtained from robust pruning methods. Furthermore, GDWS easily scales to large problem sizes since it operates on pre-trained models and doesn't require any additional training.

**Our contributions**:

1. We propose GDWS, a novel convolutional structure that can be seamlessly mapped onto off-the-shelf hardware and accelerate pre-trained CNNs significantly while maintaining robust accuracy.

2. We show that the error-optimal and complexity-optimal GDWS approximations of any pre-trained standard 2D convolution can be obtained via greedy polynomial time algorithms, thus eliminating the need for any expensive training.

3. We apply GDWS to a variety of networks on CIFAR-10, SVHN, and ImageNet to simultaneously achieve higher robustness *and* higher FPS than existing robust complexity reduction techniques, while incurring no extra training cost.

4. We demonstrate the versatility of GDWS by using it to design efficient CNNs that are robust to union of $(\ell_\infty, \ell_2, \ell_1)$ perturbation models. To the best of our knowledge, this is the first work that proposes efficient and robust networks to the union of norm-bounded perturbation models.

## 2 Background and Related Work

The problem of designing **efficient** and **robust** CNNs, though crucial for safety-critical Edge applications, is not yet well understood. Very few recent works have addressed this problem [19, 35, 42, 6, 34, 7]. We cluster prior works into the following categories:

**Quantization** Reducing the complexity of CNNs via model quantization in the absence of any adversary is a well studied problem in the deep learning literature [30, 31, 2, 44, 14, 26, 3]. The role of quantization on adversarial robustness was studied in Defensive Quantization (DQ) [19] where it was observed that conventional *post-training* fixed-point quantization makes networks more *vulnerable* to adversarial perturbations than their full-precision counterparts. EMPIR [35] also leverages extreme model quantization (up to 2-bits) to build an ensemble of efficient and robust networks. However, [40] broke EMPIR by constructing attacks that fully leverage the model structure, i.e., adaptive attacks. In contrast, GDWS is an orthogonal complexity reduction technique that preserves the base model's adversarial robustness and can be applied in conjunction with model quantization.

**Pruning** The goal of pruning is to compress neural networks by zeroing out unimportant weights [10, 8, 46, 41]. The structured pruning method in [42] combines the alternating direction method of multipliers (ADMM) [46] for parameter pruning within the AT framework [21] to design pruned and robust networks. The flexibility of ADMM enables it to achieve a high FPS on Jetson (as seen in Fig. 1a) but suffers from a significant drop in robustness. ATMC [6] augments the ADMM framework [42] with model quantization and matrix factorization to further boost the compression ratio. On the other hand, unstructured pruning methods such as HYDRA [34] prunes models via important score optimization [25]. However, HYDRA's high pruning ratios ($> 90\%$) doesn't translate into real-time FPS improvements on off-the-shelf hardware and often requires custom hardware design to fully leverage their capabilities [9]. GDWS is complementary to unstructured pruning methods, e.g., when applied to HYDRA, GDWS boosts the achievable FPS and achieves much higher robustness at iso-FPS when compared to structured (filter) pruning ADMM.

**Neural Architecture Search** Resource-efficient CNNs can be designed by exploiting design intuitions such as depthwise separable (DWS) convolutions [12, 32, 13, 47, 15, 38]. While neural architecture search (NAS) [48, 27] automates the process, it requires massive compute resources, e.g., thousands of GPU hours for a single network. Differentiable NAS [20] and one-shot NAS [1] drastically reduce the cost of this search. In [7], a one-shot NAS framework [1] is combined with the AT framework [21] to search for robust network architectures, called RobNets. RobNets achieve slightly higher robustness than existing networks with less storage requirements. In this work, we show that applying GDWS to existing architectures, e.g., WideResNet-28-4, achieves significantly higher FPS than RobNet, at iso-robustness and model size.

## 3 Generalized Depthwise-Separable Convolutions

In this section, we introduce GDWS convolutions and develop error-optimal and complexity-optimal GDWS approximations of standard 2D convolution. These optimal approximations are then employed to construct GDWS networks from any pre-trained robust CNN built from standard 2D convolutions.

**Notation:** A $(C, K, M)$ standard 2D convolution operates on an input feature map $\mathsf{X} \in \mathbb{R}^{C \times H \times W}$ via $M$ filters (also referred to as kernels or output channels) each consisting of $C$ channels each of dimension $K \times K$ to generate an output feature map $\mathsf{Y} \in \mathbb{R}^{M \times H' \times W'}$.

**2D Convolution as Matrix Multiplication:** The $M$ filters can be viewed as vectors $\{\mathbf{w}_i\}_{i=1}^{M} \in \mathbb{R}^{CK^2}$ obtained by vectorizing the $K^2$ elements within a channel and then across the $C$ channels. The resulting weight matrix $\mathbf{W} \in \mathbb{R}^{M \times CK^2}$ is constructed by stacking these filter vectors, i.e., $\mathbf{W} = [\mathbf{w}_1 | \mathbf{w}_2 | ... | \mathbf{w}_M]^{\mathrm{T}}$.

From an operational viewpoint, the matrix $\mathbf{W}$ can be used to compute the 2D convolution via Matrix Multiplication (MM) with the input matrix $\mathbf{X} = \Psi(\mathsf{X}) \in \mathbb{R}^{CK^2 \times H'W'}$:

$$\mathbf{Y} = \mathbf{W}\mathbf{X} = \mathbf{W}\Psi(\mathsf{X}) \tag{1}$$

where $\Psi$ is an unrolling operator that generates all $H'W'$ input feature map slices and stacks them in matrix format. The resultant output matrix $\mathbf{Y} \in \mathbb{R}^{M \times H'W'}$ can be reshaped via the operator $\Phi$ to retrieve $\mathsf{Y} = \Phi(\mathbf{Y})$. The computational complexity of (1) in terms of multiply-accumulate (MAC) operations is given by:

$$H'W'MCK^2 \tag{2}$$

The reshaping operators $\Phi$ and $\Psi$ are only for notational convenience and are computation-free.

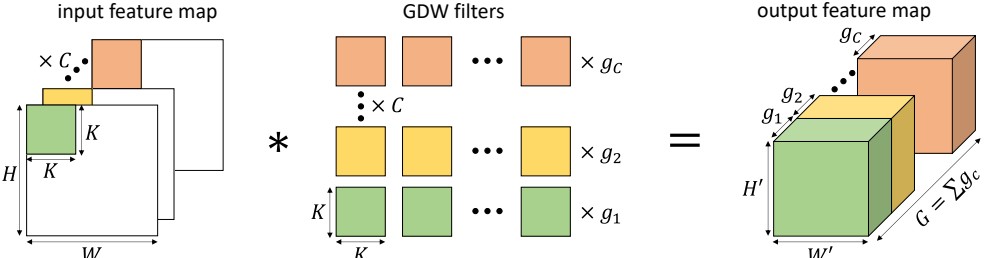

Figure 2: The $(C, K, \mathbf{g})$ generalized depthwise (GDW) convolution operation. A standard depthwise (DW) convolution is obtained by setting $g_c = 1 \ \forall c \in [C]$.

## 3.1 GDWS Formulation

**Definition:** A $(C, K, \mathbf{g}, M)$ GDWS convolution is parameterized by the *channel distribution vector* $\mathbf{g} \in \mathbb{Z}_+^C$ in addition to the parameters $(C, K, M)$ of a standard 2D convolution. A GDWS convolution is composed of a $(C, K, \mathbf{g})$ *Generalized Depthwise* (GDW) convolution and a $(G, 1, M)$ standard pointwise (PW) convolution where $G = \sum g_c$ with $c \in [C]$[1].

A $(C, K, \mathbf{g})$ GDW convolutional layer (Fig. 2) operates on an input feature map $\mathsf{X} \in \mathbb{R}^{C \times H \times W}$ by convolving the $c^{\text{th}}$ channel with $g_c \in \mathbb{Z}_+$ depthwise $K \times K$ filters to produce a total of $G$ intermediate output channels each of size $H' \times W'$. The $(G, 1, M)$ PW layer operates on the intermediate output feature map by convolving it with $M$ filters of size $1 \times 1$, thus producing the output feature map $\mathsf{Y} \in \mathbb{R}^{M \times H' \times W'}$.

**Relation to DWS:** Setting $g_c = 1 \ \forall c \in [C]$ reduces the GDWS convolution to the standard DWS convolution popularized by [12]. Thus, GDWS generalizes DWS by allowing for more than one ($g_c \geq 1$) depthwise filters per channel. This simple generalization relaxes DWS's highly constrained structure enabling accurate approximations of the 2D convolution. Thus, GDWS when applied to pre-trained models preserves its original behavior and therefore its natural and robust accuracy. Furthermore, GDWS achieves high throughput since it exploits the same hardware features that enable networks with DWS to be implemented efficiently. One might ask: why not use DWS on pre-trained models? Doing so will result in very high approximation errors. In fact, in Section 4.2, we show that applying GDWS to a pre-trained complex network such as ResNet-18 achieves better robust accuracy than MobileNet trained from scratch, while achieving similar FPS.

**GDWS Complexity:** The total number of MAC operations required by GDWS convolutions is:

$$H'W'\Big( \sum_{c=1}^{C} g_c(K^2 + M) \Big) = H'W'G(K^2 + M) \tag{3}$$

Thus, replacing standard 2D convolutions with GDWS convolutions results in a complexity reduction by a factor of $\frac{G(K^2+M)}{CK^2M}$.

## 3.2 Properties of GDWS Convolutions

We present properties of the GDWS weight matrix $\mathbf{W}$ that will be vital for developing the optimal approximation procedures.

**Property 1.** *The weight matrix $\mathbf{W} \in \mathbb{R}^{M \times CK^2}$ of a $(C, K, \mathbf{g}, M)$ GDWS convolution can be expressed as:*

$$\mathbf{W} = \mathbf{W}_{\mathrm{P}} \mathbf{W}_{\mathrm{D}} \tag{4}$$

*where $\mathbf{W}_{\mathrm{P}} \in \mathbb{R}^{M \times G}$ and $\mathbf{W}_{\mathrm{D}} \in \mathbb{R}^{G \times CK^2}$ are the weight matrices of the PW and GDW convolutions, respectively.*

Property 1 implies that any GDWS convolution has an equivalent 2D convolution whose weight matrix is the product of $\mathbf{W}_{\mathrm{P}}$ and $\mathbf{W}_{\mathrm{D}}$, where $\mathbf{W}_{\mathrm{P}}$ is a regular convolution weight matrix with $K = 1$ and $\mathbf{W}_{\mathrm{D}}$ has the following property:

---

[1] we use the notation $[C] = \{1, 2, ..., C\}$ for brevity.

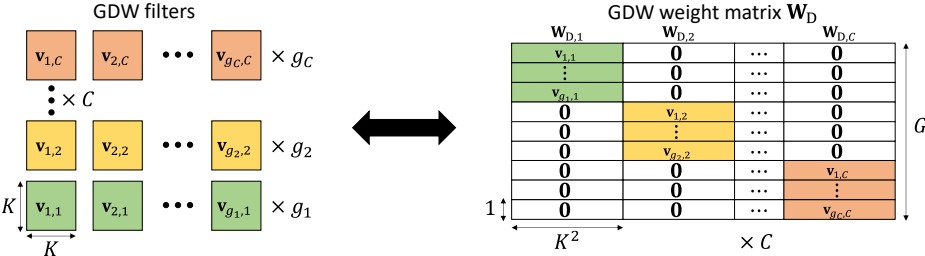

Figure 3: The weight matrix representation of a GDW convolution operation. All the $\mathbf{v}_{i,j}$ vectors are row vectors with $K^2$ elements.

**Property 2.** *The weight matrix $\mathbf{W}_{\mathrm{D}} \in \mathbb{R}^{G \times CK^2}$ of a $(C, K, \mathbf{g})$ GDW convolution has a block-diagonal structure. Specifically, $\mathbf{W}_{\mathrm{D}}$ is a concatenation of $C$ sub-matrices where each sub-matrix $\mathbf{W}_{\mathrm{D},c} \in \mathbb{R}^{G \times K^2}$ has at most $g_c$ non-zero consecutive rows, starting at row index $1 + \sum_{k=1}^{c-1} g_k$ as shown in Fig. 3.*

This structure is due to the fact that input channels are convolved independently with at most $g_c$ depthwise filters per channel. Finally, let $\mathbf{W} = [\mathbf{W}_1 | \mathbf{W}_2 | ... | \mathbf{W}_C]$ be represented as the concatenation of $C$ sub-matrices, then combining Properties 1 & 2 establishes the following lemma:

**Lemma 1.** *The weight matrix of a $(C, K, \mathbf{g}, M)$ GDWS convolution can be expressed as the concatenation of $C$ sub-matrices $\mathbf{W}_c$ where $\mathrm{rank}(\mathbf{W}_c) \leq \min(g_c, K^2) \; \forall c \in [C]$.*

The reason for this is that each sub-matrix $\mathbf{W}_c$ can be expressed as the sum of $g_c$ rank 1 matrices of size $M \times K^2$. A detailed proof of Lemma 1 can be found in the Appendix. A major implication of Lemma 1 is that any 2D standard convolution is equivalent to a GDWS convolution with $g_c = K^2$ $\forall c$[2]. Hence, in the rest of this paper we will assume $g_c \leq K^2$ when we approximate 2D convolutions with GDWS.

### 3.3 Optimal GDWS Approximation Methods

We wish to approximate a standard 2D convolution with weight matrix $\mathbf{W} \in \mathbb{R}^{M \times CK^2}$ with a GDWS convolution with weight matrix $\mathbf{Q} \in \mathbb{R}^{M \times CK^2}$ to minimize the weighted approximation error defined as:

$$e(\mathbf{W}, \mathbf{Q}, \boldsymbol{\alpha}) = \sqrt{\sum_{c=1}^{C} \alpha_c ||\mathbf{W}_c - \mathbf{Q}_c||_{\mathrm{F}}^2} \tag{5}$$

where $||.||_{\mathrm{F}}$ denotes the Frobenius norm of a matrix and $\boldsymbol{\alpha} \in \mathbb{R}_{+}^{C}$ is a vector of positive weights. Setting $\alpha_c = 1 \; \forall c$ simplifies (5) to the Frobenius norm of the error matrix $\mathbf{W} - \mathbf{Q}$. Furthermore, from (3), one can upper bound the complexity of the GDWS approximation $\mathbf{Q}$ via an upper bound on $G = \sum g_c$ where the $g_c$'s are obtained from Lemma 1.

Based on the GDWS properties and Lemma 1, we state the following error-optimal approximation theorem:

**Theorem 1.** *Given a $(C, K, M)$ standard 2D convolution with weight matrix $\mathbf{W}$, the $(C, K, \mathbf{g}, M)$ GDWS approximation with weight matrix $\hat{\mathbf{W}}$ that minimizes the error in (5) subject to $\sum g_c = G \leq \gamma$ (for some $\gamma \in \mathbb{Z}_{+}$), can be obtained in polynomial time via Algorithm 1.*

That is:

$$\hat{\mathbf{W}} = \underset{\mathbf{Q}: \, G \leq \gamma}{\arg\min} \, e(\mathbf{W}, \mathbf{Q}, \boldsymbol{\alpha}) \tag{6}$$

can be solved for any weight error vector $\boldsymbol{\alpha} \in \mathbb{R}_{+}^{C}$ in polynomial time. While Theorem 1 shows that the optimal GDWS approximation under a complexity constraint can be solved efficiently, a similar result can be obtained for the reverse setting shown next.

---

[2] Typical CNNs satisfy $K^2 < M$, e.g., $K = 3$ and $M \geq 16$

**Theorem 2.** *Given a $(C, K, M)$ standard 2D convolution with weight matrix $\mathbf{W}$, the $(C, K, \mathbf{g}, M)$ GDWS approximation with weight matrix $\hat{\mathbf{W}}$ that minimizes the complexity in* (3) *subject to $e(\mathbf{W}, \mathbf{Q}, \boldsymbol{\alpha}) \leq \beta$ (for some $\beta \geq 0$), can be constructed in polynomial time via Algorithm* 2.

That is:

$$\hat{\mathbf{W}} = \underset{\mathbf{Q}: \, e(\mathbf{W}, \mathbf{Q}, \boldsymbol{\alpha}) \leq \beta}{\arg\min} \sum_{c=1}^{C} g_c \qquad (7)$$

can be solved for any weight error vector $\boldsymbol{\alpha} \in \mathbb{R}_+^C$ in polynomial time. Proofs of Theorems 1 & 2 can be found in the Appendix.

---

**Algorithm 1:** (MEGO) Minimum Error Complexity-constrained GDWS Optimal Approximation

---

**Input:** A $(C, K, M)$ convolution $\mathbf{W}$,
       weight error vector $\boldsymbol{\alpha}$, and
       constraint $\gamma \in \mathbb{Z}_+$.
**Output:** A $(C, K, \mathbf{g}, M)$ GDWS
       convolution $\hat{\mathbf{W}}$, satisfying
       $\sum g_c \leq \gamma$.
1   Compute SVDs of
    $\mathbf{W}_c = \sum_{i=1}^{r_c} \sigma_{i,c} \mathbf{u}_{i,c} \mathbf{v}_{i,c}^\mathsf{T}$
2   Initialize $\mathbf{g} = \mathbf{0}$
3   **while** $\sum g_c < \gamma$ **do**
4     |   $c' = \arg\max_c \alpha_c \sigma_{g_c+1,c}^2$    // $g_c < r_c$
5     |   $g_{c'} \leftarrow g_{c'} + 1$
6   Compute $\hat{\mathbf{W}}_c$ via truncated SVD of $\mathbf{W}_c$
    with rank $g_c$: $\hat{\mathbf{W}}_c = \sum_{i=1}^{g_c} \sigma_{i,c} \mathbf{u}_{i,c} \mathbf{v}_{i,c}^\mathsf{T}$
7   Construct $\hat{\mathbf{W}} = [\hat{\mathbf{W}}_1 | ... | \hat{\mathbf{W}}_C]$

---

**Algorithm 2:** (LEGO) Least Complex Error-constrained GDWS Optimal Approximation

---

**Input:** A $(C, K, M)$ convolution $\mathbf{W}$,
       weight error vector $\boldsymbol{\alpha}$, and constraint
       $\beta \geq 0$.
**Output:** A $(C, K, \mathbf{g}, M)$ GDWS
       convolution $\hat{\mathbf{W}}$, satisfying $e \leq \beta$.
1   Compute SVDs of $\mathbf{W}_c = \sum_{i=1}^{r_c} \sigma_{i,c} \mathbf{u}_{i,c} \mathbf{v}_{i,c}^\mathsf{T}$
2   Initialize $g_c = r_c$, $b = 0$,
    $c' = \arg\min_c \alpha_c \sigma_{r_c,c}^2$, $h = \alpha_{c'} \sigma_{r_{c'},c'}^2$
3   **while** $b + h < \beta$ **do**
4     |   $b \leftarrow b + h$ and $g_{c'} \leftarrow g_{c'} - 1$
5     |   $c' = \arg\min_c \alpha_c \sigma_{g_c,c}^2$    // $g_c > 1$
6     |   $h = \alpha_{c'} \sigma_{r_{c'},c'}^2$
7   Compute $\hat{\mathbf{W}}_c$ via truncated SVD of $\mathbf{W}_c$
    with rank $g_c$: $\hat{\mathbf{W}}_c = \sum_{i=1}^{g_c} \sigma_{i,c} \mathbf{u}_{i,c} \mathbf{v}_{i,c}^\mathsf{T}$
8   Construct $\hat{\mathbf{W}} = [\hat{\mathbf{W}}_1 | ... | \hat{\mathbf{W}}_C]$

---

### 3.4 Constructing GDWS Networks

When confronted with a CNN with $L$ convolutional layers, the question arises: *How to assign resources (in terms of complexity) amongst the $L$ layers such that the robustness of the CNN is minimally compromised?* To answer this question, we compute per-layer weight error vectors $\boldsymbol{\alpha}_l$, such that the computed error in (5) weighs how different sub-matrices affect the final output of the CNN.

Let $f : \mathbb{R}^D \to \mathbb{R}^N$ be a pre-trained CNN for an $N$-way classification problem with $L$ convolutional layers parameterizd by weight matrices $\mathbf{W}^{(l)} \in \mathbb{R}^{M_l \times C_l K_l^2}$. The CNN $f$ operates on a $D$-dimensional input vector $\mathbf{x}$ to produce a vector $\mathbf{z} = f(\mathbf{x})$ of soft outputs or logits. Denote by $n_x \in [N]$ the predicted class label associated with $\mathbf{x}$, and define $\delta_{x,j} = z_j - z_{n_x}$ to be the soft output differences $\forall j \in [N] \setminus \{n_x\}$.

Inspired by [30, 31], we propose a simple yet effective method for computing the per-layer weight error vectors as follows:

$$\alpha_{c,l} = \frac{1}{M_l K_l^2} \mathbb{E}\left[ \sum_{\substack{j=1 \\ j \neq n_x}}^{N} \frac{||\mathbf{D}_{x,j}^{(c,l)}||_\mathrm{F}^2}{2\delta_{x,j}^2} \right] \qquad \forall l \in [L], \, \forall c \in [C_l] \qquad (8)$$

where $\mathbf{D}_{x,j}^{(c,l)} \in \mathbb{R}^{M_l \times K_l^2}$ is the derivative of $\delta_{x,j}$ w.r.t. the sub-matrix $\mathbf{W}_c^{(l)}$. The expectation is taken over the input vector $\mathbf{x}$. Equation (8) can be thought of as the expected noise gain from a particular channel in a particular layer to the network output required to flip its decision. The Appendix provides a detailed rationale underlying (8).

Computation of (8) can be simplified by obtaining an estimate of the mean over a small batch of inputs sampled from the training set and by leveraging software frameworks such as PyTorch

[24] that automatically take care of computing $\mathbf{D}_{x,j}^{(c,l)}$. Algorithm 3 summarizes the steps required to approximate any pre-trained CNN with an equivalent CNN utilizing GDWS convolutions. Unless specified otherwise, all the results in this paper are obtained via Algorithm 3.

---

**Algorithm 3:** Constructing GDWS networks

**Input:** CNN $f$ with convolutional layers $\{\mathbf{W}^{(l)}\}$, $\{\boldsymbol{\alpha}_l\}$ computed via (8), and constraint $\beta \geq 0$.

**Output:** CNN $\hat{f}$ with GDWS convolutions $\{\hat{\mathbf{W}}^{(l)}\}$

1  $\hat{f} = f$                                                             // Initialize $\hat{f}$

2  **for** $l \in \{1, ..., L\}$ **do**

3      $\hat{\mathbf{W}}^{(l)} = \texttt{LEGO}(\mathbf{W}^{(l)}, \boldsymbol{\alpha}_l, \beta)$                         // solve via Algorithm 2

4      Decompose $\hat{\mathbf{W}}^{(l)}$ into GDW and PW convolutions via Property 1 and Lemma 1

5      Replace the $l^{\text{th}}$ convolution layer in $\hat{f}$ with GDW and PW convolutions

---

## 4 Experiments

### 4.1 Evaluation Setup

We measure the throughput in FPS by mapping the networks onto an NVIDIA Jetson Xavier via native PyTorch [24] commands. We experiment with VGG-16 [36], ResNet-18[3] [11], ResNet-50, and WideResNet-28-4 [43] network architectures, and report both natural accuracy ($\mathcal{A}_{\text{nat}}$) and robust accuracy ($\mathcal{A}_{\text{rob}}$). Following standard procedure, we report $\mathcal{A}_{\text{rob}}$ against $\ell_\infty$ bounded perturbations generated via PGD [21] with standard attack strengths: $\epsilon = 8/255$ with PGD-100 for both CIFAR-10 [17] and SVHN [23] datasets, and $\epsilon = 4/255$ with PGD-50 for the ImageNet [29] dataset. Section 4.3 studies union of multiple perturbation models ($\ell_\infty, \ell_2, \ell_1$). In the absence of publicly released pre-trained models, we establish strong baselines using AT [21] following the approach of [28] which utilizes early stopping to avoid robust over-fitting. Details on the training/evaluation setup can be found in the Appendix.

Table 1: Comparison between RobNet [7] and GDWS on the CIFAR-10 dataset. GDWS is applied to standard pre-trained models.

| Models | $\mathcal{A}_{\text{nat}}$ [%] | $\mathcal{A}_{\text{rob}}$ [%] | Size [MB] | FPS |
|---|---|---|---|---|
| RobNet [7] | 82.72 | 52.23 | 20.8 | 5 |
| ResNet-50 | 84.21 | 53.05 | 89.7 | 16 |
| + GDWS ($\beta = 0.001$) | **83.72** | **52.94** | 81.9 | **37** |
| WRN-28-4 | 84.00 | 51.80 | 22.3 | 17 |
| + GDWS ($\beta = 1 \times 10^{-5}$) | **83.27** | **51.70** | 18.9 | **65** |
| ResNet-18 | 82.41 | 51.55 | 42.6 | 28 |
| + GDWS ($\beta = 0.005$) | **81.17** | **50.98** | 29.1 | **104** |
| VGG-16 | 77.49 | 48.92 | 56.2 | 36 |
| + GDWS ($\beta = 0.25$) | **77.17** | **49.56** | 28.7 | **129** |

### 4.2 Results

**Ablation Study:** We first show the effectiveness of GDWS on the CIFAR-10 datasets using four network architectures. Table 1 summarizes $\mathcal{A}_{\text{nat}}$ and $\mathcal{A}_{\text{rob}}$ as well as FPS and model size. It is clear that GDWS networks preserve robustness as both $\mathcal{A}_{\text{nat}}$ and $\mathcal{A}_{\text{rob}}$ are always within $\sim 1\%$ of their respective baselines. The striking conclusion is that in spite of GDWS offering modest reductions in model size, it *drastically* improves the FPS of the base network across diverse architectures. For instance, a ResNet-18 utilizing GDWS convolutions is able to run at 104 FPS compared to the baseline's 28 FPS (>250% improvement) without additional training and without compromising on robust accuracy. In the Appendix, we explore the benefits of applying GDWS using both Algorithms 1 & 2, provide more detailed results on CIFAR-10 and show that similar gains are observed with SVHN dataset.

---

[3]For CIFAR-10 and SVHN, we use the standard pre-activation version of ResNets.

**GDWS vs. RobNet:** In Table 1, we also compare GDWS networks (obtained from standard networks) with a publicly available pre-trained RobNet model, the robust network architecture designed via the NAS framework in [7]. Note that RobNet utilizes DWS convolutions which precludes the use of GDWS. However, despite the efficiency of DWS convolutions in RobNet, its irregular cell structure leads to extremely poor mapping on the Jetson as seen by its low 5 FPS. For reference, a standard WideResNet-28-4 (WRN-28-4) runs at 17 FPS with similar robustness and model size. Applying GDWS to the WideResNet-28-4 further increases the throughput to 65 FPS which is a 1200% improvement compared to RobNet while maintaining robustness. This further supports our assertion that model compression alone does not lead to enhanced performance on real hardware.

**GDWS vs. Lightweight Networks:**
A natural question that might arise from this work: why not train lightweight networks utilizing DWS convolutions from scratch instead of approximating pre-traind complex networks with GDWS? In Table 2, we compare the performance of GDWS networks (obtained from Table 1) vs. standard lightweight networks: MobileNetV1 [12], MobileNetV2 [32], and ResNet-20 [11], as well as a DWS-version of the standard ResNet-18 trained from scratch on the CIFAR-

Table 2: Comparison between GDWS and lightweight networks on the CIFAR-10 dataset. The GDWS numbers are from Table 1.

| Models | $\mathcal{A}_{nat}$ [%] | $\mathcal{A}_{rob}$ [%] | Size [MB] | FPS |
|---|---|---|---|---|
| ResNet-18 + GDWS | **81.17** | **50.98** | 29.1 | **104** |
| VGG-16 + GDWS | 77.17 | 49.56 | 28.7 | 129 |
| MobileNetV1 | 79.92 | 49.08 | 12.3 | 125 |
| MobileNetV2 | 79.59 | 48.55 | 8.5 | 70 |
| ResNet-18 (DWS) | 80.12 | 48.52 | 5.5 | 120 |
| ResNet-20 | 74.82 | 47.00 | 6.4 | 125 |

10 dataset. We find that applying GDWS to a pre-trained complex network such as ResNet-18 achieves better $\mathcal{A}_{nat}$ and $\mathcal{A}_{rob}$ than all lightweight networks, while achieving DWS-like FPS and requiring no extra training despite offering modest reductions in model size. The only benefit of using lightweight networks is the much smaller model size compared to GDWS networks.

**GDWS vs. Structured Pruning:** In Table 3, we compare GDWS with the robust structured pruning method ADMM [42] on CIFAR-10, using two networks: VGG-16 and ResNet-18. Due to the lack of publicly available pre-trained models, we use their released code to reproduce both the AT baselines, and the corresponding pruned models at different pruning ratios. The nature of structured pruning allows ADMM pruned networks ($p \geq$ 50%) to achieve both high compression ratios and significant improvement in FPS over their un-pruned baselines but at the expense of robust-

Table 3: Comparison between ADMM [42] and GDWS using VGG-16 and ResNet-18 on CIFAR-10.

| Models | $\mathcal{A}_{nat}$ [%] | $\mathcal{A}_{rob}$ [%] | Size [MB] | FPS |
|---|---|---|---|---|
| VGG-16 (AT from [42]) | 77.45 | 45.78 | 56.2 | 36 |
| + GDWS ($\beta = 0.5$) | **76.40** | **46.28** | 38.8 | **119** |
| VGG-16 ($p = 25\%$) | 77.88 | 43.80 | 31.6 | 26 |
| VGG-16 ($p = 50\%$) | 75.33 | 42.93 | 14.0 | 113 |
| VGG-16 ($p = 75\%$) | 70.39 | 41.07 | 3.5 | 174 |
| ResNet-18 (AT from [42]) | 80.65 | 47.05 | 42.6 | 28 |
| + GDWS ($\beta = 0.75$) | **79.13** | **46.15** | 30.4 | **105** |
| ResNet-18 ($p = 25\%$) | 81.61 | 42.67 | 32.1 | 31 |
| ResNet-18 ($p = 50\%$) | 79.42 | 42.23 | 21.7 | 60 |
| ResNet-18 ($p = 75\%$) | 74.62 | 43.23 | 11.2 | 74 |

ness and accuracy. For instance, a ResNet-18 with 75% of its channels pruned results in a massive 7% (4%) drop in $\mathcal{A}_{nat}$ ($\mathcal{A}_{rob}$) compared to the baseline even though it achieves a 160% improvement in FPS. In contrast, a post-training application of GDWS to the *same* ResNet-18 baseline results in a massive 275% improvement in FPS while preserving both $\mathcal{A}_{nat}$ and $\mathcal{A}_{rob}$ within 1% of their baseline values. Thus, despite achieving modest compression ratios compared to ADMM, GDWS achieves comparable improvements in FPS without compromising robustness.

**GDWS vs. Unstructured Pruning:** We compare GDWS with HYDRA [34] which is an unstructured robust pruning method, on both CIFAR-10 and ImageNet datasets. We use the publicly released HYDRA models as well as their AT baselines, and apply GDWS to both the un-pruned and pruned models. Table 4 summarizes the robustness and FPS of HYDRA and GDWS networks on CIFAR-10. HYDRA pruned models have arbitrarily sparse weight matrices that cannot be leveraged by off-the-shelf hardware platforms immediately. Instead, we rely on the extremely high sparsity (99%) of these matrices to emulate channel pruning whereby channels are discarded only if all filter weights are zero.

This explains why, despite their high compression ratios, HYDRA models do not achieve significant improvements in FPS compared to their baselines.

For instance, a 99% HYDRA pruned WideResNet model achieves a massive $\sim100\times$ compression ratio and improves the FPS from 17 to 28, but suffers from a large $\sim10\%$ drop in both $\mathcal{A}_{\text{nat}}$ and $\mathcal{A}_{\text{rob}}$. In contrast, GDWS applied to the same un-pruned baseline preserves robustness and achieves significantly better throughput of 68 FPS, even though the model size reduction is negligible. Interestingly, we find that applying GDWS directly to HYDRA pruned models results in networks with high compression ratios with no robustness degradation and massive improvements in FPS compared to the pruned baseline. For example, applying GDWS to the same 99% HYDRA pruned WideResNet achieves a $\sim20\times$ compression ratio and improves the throughput from 28 FPS to 68 FPS while preserving $\mathcal{A}_{\text{nat}}$ and $\mathcal{A}_{\text{rob}}$ of the pruned baseline. This

Table 4: Comparison between HYDRA [34] and GDWS using VGG-16 and WRN-28-4 on CIFAR-10.

| Models | $\mathcal{A}_{\text{nat}}$ [%] | $\mathcal{A}_{\text{rob}}$ [%] | Size [MB] | FPS |
|---|---|---|---|---|
| VGG-16 (AT from [34]) | 82.72 | 51.93 | 58.4 | 36 |
| + GDWS ($\beta = 0.5$) | **82.53** | **50.96** | 50.6 | **102** |
| VGG-16 ($p = 90\%$) | 80.54 | 49.44 | 5.9 | 36 |
| + GDWS ($\beta = 0.1$) | 80.47 | 49.52 | 31.5 | 93 |
| VGG-16 ($p = 95\%$) | 78.91 | 48.74 | 3.0 | 36 |
| + GDWS ($\beta = 0.1$) | 78.71 | 48.53 | 18.3 | 106 |
| VGG-16 ($p = 99\%$) | 73.16 | 41.74 | 0.6 | 41 |
| + GDWS ($\beta = 0.02$) | **72.75** | **41.56** | **2.9** | **136** |
| WRN-28-4 (AT from [34]) | 85.35 | 57.23 | 22.3 | 17 |
| + GDWS ($\beta = 1$) | **84.17** | **55.87** | 20.5 | **68** |
| WRN-28-4 ($p = 90\%$) | 83.69 | 55.20 | 2.3 | 17 |
| + GDWS ($\beta = 0.125$) | 83.38 | 54.79 | 11.9 | 59 |
| WRN-28-4 ($p = 95\%$) | 82.68 | 54.18 | 1.1 | 17 |
| + GDWS ($\beta = 0.005$) | 82.59 | 54.22 | 7.2 | 60 |
| WRN-28-4 ($p = 99\%$) | 75.62 | 47.21 | 0.2 | 28 |
| + GDWS ($\beta = 0.0025$) | **75.36** | **47.04** | **1.2** | **68** |

synergy between HYDRA and GDWS is due to the fact that highly sparse convolution weight matrices are more likely to have low-rank and sparse sub-matrices. This implies that, using Lemma 1, sparse convolutions can be transformed to sparse GDWS versions with negligible approximation error. We explore this synergy in detail in the Appendix. Table 5 shows that GDWS benefits also show up in ImageNet using ResNet-50.

Table 5: Comparison between HYDRA [34] and GDWS using ResNet-50 on ImageNet.

| Models | top-1 / 5 $\mathcal{A}_{\text{nat}}$ [%] | top-1 / 5 $\mathcal{A}_{\text{rob}}$ [%] | Size [MB] | FPS |
|---|---|---|---|---|
| ResNet-50 (AT from [34]) | 60.25 / 82.39 | 31.94 / 61.13 | 97.5 | 15 |
| + GDWS ($\beta = 50$) | **58.04 / 80.56** | **30.22 / 58.48** | **86.2** | **19** |
| ResNet-50 ($p = 95\%$) | 44.60 / 70.12 | 19.53 / 44.28 | 5.1 | 15 |
| + GDWS ($\beta = 0.5$) | 43.91 / 69.46 | 19.27 / 43.58 | 12.6 | 19 |
| ResNet-50 ($p = 99\%$) | 27.68 / 52.55 | 11.32 / 28.83 | 1.2 | 17 |
| + GDWS ($\beta = 0.5$) | **26.27 / 50.90** | **10.92 / 27.55** | **2.9** | **25** |

## 4.3 Defending against Union of Perturbation Models

Recent work has shown that adversarial training with a single perturbation model leads to classifiers vulnerable to the union of $(\ell_\infty, \ell_2, \ell_1)$-bounded perturbations [33, 39, 22]. The method of multi steepest descent (MSD) [22] achieves state-of-the-art union robust accuracy ($\mathcal{A}_{\text{rob}}^{\text{U}}$) against the union of $(\ell_\infty, \ell_2, \ell_1)$-bounded perturbations. We demonstrate the versatility of GDWS by applying it to a publicly available [22] robust pre(MSD)-trained ResNet-18 model on CIFAR-10. Following the setup in [22], all attacks were run on a subset of the first 1000 test images with 10 random restarts with the following attack configurations: $\epsilon_\infty = 0.03$ with PGD-100 , $\epsilon_2 = 0.5$ with PGD-500, and $\epsilon_1 = 12$ with PGD-100. Table 6 shows that applying GDWS with $\beta = 0.01$ to the pre-trained ResNet-18 incurs a negligible ($< \sim 1\%$) drop in $\mathcal{A}_{\text{nat}}$ and $\mathcal{A}_{\text{rob}}^{\text{U}}$ while improving the throughput from 28 FPS to 101 FPS ($> 250\%$ improvement).

Table 6: Benefits of GDWS when evaluated against union of perturbation models on CIFAR-10. $\mathcal{A}_{\text{rob}}^{\text{U}}$ is the fraction of test images that are simultaneously resistant to all perturbation models.

| Models | $\mathcal{A}_{\text{nat}}$ [%] | $\mathcal{A}_{\text{rob}}^{\infty}$ [%] | $\mathcal{A}_{\text{rob}}^{1}$ [%] | $\mathcal{A}_{\text{rob}}^{2}$ [%] | $\mathcal{A}_{\text{rob}}^{\text{U}}$ [%] | FPS |
|---|---|---|---|---|---|---|
| ResNet-18 (AT from [22]) | 81.74 | 47.50 | 53.60 | 66.10 | 46.10 | 28 |
| + GDWS ($\beta = 0.0025$) | 81.67 | 47.60 | 53.60 | 66.00 | 46.30 | 87 |
| + GDWS ($\beta = 0.005$) | 81.43 | 47.30 | 52.60 | 65.60 | 45.70 | 92 |
| + GDWS ($\beta = 0.01$) | **81.10** | **47.20** | **52.20** | **65.00** | **45.20** | **101** |

## 5 Discussion

We have established that the proposed GDWS convolutions are universal and efficient approximations of standard 2D convolutions that are able to accelerate any pre-trained CNN utilizing standard 2D convolution while preserving its accuracy and robustness. This facilitates the deployment of CNNs in safety critical edge applications where real-time decision making is crucial and robustness cannot be compromised. One limitation of this work is that GDWS alone does not achieve high compression ratios compared to pruning. Combining unstructured pruning with GDWS alleviates this problem to some extent. Furthermore, GDWS cannot be applied to CNNs utilizing DWS convolutions, such as RobNet for instance. An interesting question is to explore the possibility of training GDWS-structured networks from scratch. Another possible direction is fine-tuning post GDWS approximation to recover robustness, which we explore in the Appendix.

In summary, a GDWS approximated network inherits all the properties, e.g., accuracy, robustness, compression and others, of the baseline CNN while significantly enhancing its throughput (FPS) on real hardware. Therefore, the societal impact of GDWS approximated networks are also inherited from those of the baseline CNNs.

## Acknowledgments

This work was supported by the Center for Brain-Inspired Computing (C-BRIC) and the Artificial Intelligence Hardware (AIHW) program funded by the Semiconductor Research Corporation (SRC) and the Defense Advanced Research Projects Agency (DARPA).

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
