# Supplementary Material for "Generalized Depthwise-Separable Convolutions for Adversarially Robust and Efficient Neural Networks"

## Contents

# 1 Experimental Setup Details

## 1.1 Evaluation Setup

In this section we provide details on how we measure FPS on the Jetson, as well as explain how we map GDWS convolutions efficiently. We use a single off-the-shelf NVIDIA Jetson Xavier NX developer kit for all our experiments. The Jetson Xavier is equipped with a 384-core NVIDIA Volta GPU, a 6-core NVIDIA Carmel ARM 64-bit CPU, and 8GB 128-bit LPDDR4x memory. We install the latest PyTorch packages onto the Jetson, as we will use their native neural network (NN) modules to implement both standard and GDWS convolutions. Specifically, we used PyTorch v1.8.0 with Python v3.6.9 and CUDA v10.2.

**Measuring FPS:** The Python pseudo-code in 1 explains how the FPS for any neural network model was measured on the Jetson. The main idea is to run successive inferences (batch size of 1) and measure the total elapsed time reliably, and calculate the FPS as the total number of inferences divided by the total elapsed time. To ensure consistency, we use 10000 inferences to measure FPS, after the GPU has been warmed up with 5000 inferences as well. Note that the measured FPS reflects the raw capabilities of the GPU, ignoring any I/O to and from the GPU.

```python
#get the appropriate NN architecture, e.g., ResNet-18
model =  get_architecture()

#load the pre-trained model parameters from memory
model.load_state_dict(state_dict)

#transfer the model onto the GPU
model = model.cuda()

#set the model in evaluation mode
model.eval()

#sample a single test input and load it into GPU memory
x_test = get_input() #use batch size of 1
x_test.cuda()

#ensure no gradient overheads are introduced
with torch.no_grad():
    ## run successive inferences to warm-up the GPU
    for t in range(5000):
        y = model(x_test)

    ## setup synchronized timers in PyTorch
    start = torch.cuda.Event(enable_timing=True)
    end = torch.cuda.Event(enable_timing=True)
    start.record()

    ## now that the GPU is warmed-up, we run successive inferences and
     measure the total latency
    for t in range(num_inferences):
        y = model(x_test)

    ## measure the elapsed time delay in seconds
    end.record()
    torch.cuda.synchronize()
    delay = start.elapsed_time(end)/1000

## get the frames-per-second number
FPS = num_inferences/delay
```

Listing 1: Python pseudo-code for measuring FPS on the Jetson using PyTorch modules.

**Mapping GDWS Convolutions:** Mapping GDWS convolutions requires mapping both the GDW and the PW convolutions efficiently onto the Jetson. PW layers are standard 2D convolutions with

$1 \times 1$ kernels, thus implementing PW convolutions using the PyTorch convolution module is straight forward. The challenge arises when mapping GDW convolutions, as it is a new convolutional structure that is not directly supported yet in PyTorch. To that end, we use simple tensor manipulations and leverage the existing support for standard DW convolution in PyTorch to implement GDW convolutions.

Note that a $(C, K, \mathbf{g})$ GDW convolution operating on input tensor $\mathsf{X} \in \mathbb{R}^{C \times H \times W}$ convolves the $c^{\text{th}}$ input channel with $g_c \in \mathbb{Z}_+$ depthwise $K \times K$ filters to produce a total of $G$ intermediate output channels. A DW convolution operating on the same input tensor $\mathsf{X}$ is a special case of GDW where $g_c = 1 \ \forall c$. It is not difficult to see that a $(C, K, \mathbf{g})$ GDW convolution operating on $\mathsf{X}$ is equivalent to a DW convolution operating on the modified tensor $\mathsf{X}' \in \mathbb{R}^{G \times H \times W}$ with $G = \sum g_c$ channels, where the tensor $\mathsf{X}'$ is obtained by duplicating the $c^{\text{th}}$ channel from $\mathsf{X}$ $g_c$ times. This tensor manipulation is implemented via simple tensor indexing in PyTorch. Therefore, we can efficiently map GDWS convolutions onto the Jetson without requiring any custom libraries.

## 1.2   Training Hyperparameters

In the absence of any publicly available pre-trained models, we obtain strong baselines using AT [2] following the approach of [4] which utilizes early stopping to avoid robust over-fitting. We use the same hyperparameters, detailed below for our CIFAR-10 and SVHN baselines. A single workstation with two NVIDIA Tesla P100 GPUs is used for running all the training experiments.

**CIFAR-10:** For the CIFAR-10 experiments presented in Table 3 (Table 1 in main manuscript), we use PGD-7 adversarial training with $\epsilon = 8/255$ and step size $2/255$ for a maximum of 200 epochs and 128 mini-batch size. We employ a step-wise learning rate decay set initially at 0.1 and divided by 10 at epochs 100 and 150. We use a weight decay of $5 \times 10^{-4}$, except for the lightweight networks which were trained with a smaller weight decay of $2 \times 10^{-4}$.

**SVHN:** For the SVHN experiments presented in Table 4, we use PGD-7 adversarial training with $\epsilon = 8/255$ and step size $2/255$ for a maximum of 200 epochs and 128 mini-batch size. We employ a step-wise learning rate decay set initially at 0.01 and divided by 10 at epochs 100 and 150. We use a weight decay of $5 \times 10^{-4}$.

## 1.3   Computing the Weight Error Vectors

Constructing GDWS networks via Algorithm 3 requires computing the per-layer weight error vectors $\{\boldsymbol{\alpha}_l\}$ as described in (8) in Section 3.4 of the main manuscript. Throughout all of our experiments, we compute the $\{\boldsymbol{\alpha}_l\}$ via an estimate of the mean over a small batch of *adversarial* inputs sampled from the training set. Specifically, throughout all of experiments, we use 1000 input samples generated via PGD-7 with $\epsilon = 8/255$, except for ImageNet were 5000 adversarial input samples were used that were generated via PGD-4 with $\epsilon = 4/255$.

## 2 Additional Experiments and Comparisons

### 2.1 Extended Ablation Study

**Benefits of Non-uniform GDWS Networks:** We expand on Section 4.2 by comparing the benefits of using Algorithm 3 vs. Algorithm 1 to design GDWS networks. We denote networks obtained from Algorithm 3 as GDWS-N (non-uniform reduction in complexity) and Algorithm 1 as GDWS-U (uniform reduction in complexity). Specifically, for GDWS-U, we use Algorithm 1, with unweighted error ($\alpha_{c,l} = 1$) to construct the error-optimal GDWS approximations of each layer, such that we reduce the number of MACs of each layer by the same fixed percentage.

We use VGG-16 on CIFAR-10 as our network and dataset of choice. We obtain different GDWS networks by varying the choice of $\beta$ in GDWS-N and the reduction percentage in GDWS-U. Figure 1a shows the per-layer reduction in MACs for both methods. As expected, GDWS-U produces uniform reductions across all layers, whereas GDWS-N is not restricted in that regard. In Figs 1b & 1c we compare both methods by plotting the natural and robust accuracies vs. FPS, respectively. The per-layer granularity inherit to GDWS-N allows it to outperform GDWS-U, as it consistently achieves higher natural and robust accuracies than at iso-FPS.

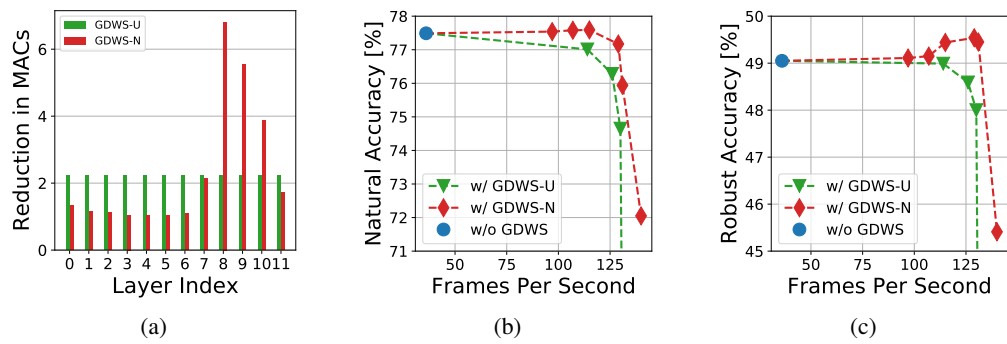

|       |       |       |
|:-----:|:-----:|:-----:|
| (a)   | (b)   | (c)   |

Figure 1: Comparison of two methods for constructing GDWS networks using VGG-16 on CIFAR-10 by showing: (a) the per-layer reduction in complexity, (b) natural accuracy vs. FPS, and (c) robust accuracy vs. FPS.

**Impact of Fine-tuning:** In this section, we showcase that fine-tuning via adversarial training for 10 epochs after the application of GDWS can significantly boost the efficacy of GDWS. In Table 1, we use the same VGG-16 baseline on CIFAR-10 from Table 1 in Section 4.2 of the main manuscript and apply GDWS with higher approximation errors $\beta$. This results in GDWS networks with smaller model sizes and higher FPS, but with a significant degradation in robust and natural accuracies. As expected, fine-tuning boosts both robust and natural accuracies (up to $\sim 1\%$ of the pre-trained baseline).

Table 1: Fine-tuning after GDWS using VGG-16 on CIFAR-10.

| Models | $\mathcal{A}_{\mathbf{nat}}$ [%] | $\mathcal{A}_{\mathbf{rob}}$ [%] | Size [MB] | FPS |
|:-------|:-----:|:-----:|:-----:|:---:|
| VGG-16 | 77.49 | 48.92 | 56.2 | 36 |
| + GDWS ($\beta = 0.25$) | 77.17 | 49.56 | 28.7 | 129 |
| + GDWS ($\beta = 2$) | 72.05 | 45.35 | 19.1 | 140 |
|   + fine-tune | **77.15** | **47.87** | 19.1 | 140 |
| + GDWS ($\beta = 5$) | 63.21 | 37.78 | 16.3 | 143 |
|   + fine-tune | **76.76** | **47.92** | 16.3 | 143 |

**Different Types of Attacks:** In this section, we conduct an extra set of attacks, highlighted in Table 2 below, on the VGG-16 network on CIFAR-10 (same baseline as before). We use the Foolbox [3] (`https://github.com/bethgelab/foolbox`) implementation of all these attacks to ensure proper

implementation. All the attacks are using $\ell_\infty$-bounded perturbations with $\epsilon = 8/255$, similar to our PGD results in the main manuscript. As expected, GDWS preserves the robustness of the pre-trained baseline, across different attack methods.

Table 2: Robustness across different types of attacks using VGG-16 on CIFAR-10.

| Models | $\mathcal{A}_{\mathbf{rob}}$ [%] (FGSM) | $\mathcal{A}_{\mathbf{rob}}$ [%] (BIM) | $\mathcal{A}_{\mathbf{rob}}$ [%] (DeepFool) | FPS |
|---|---|---|---|---|
| VGG-16 | 52.53 | 49.61 | 47.89 | 36 |
| + GDWS ($\beta = 0.25$) | 53.19 | 50.08 | 47.28 | 129 |
| + GDWS ($\beta = 0.5$) | 52.69 | 49.87 | 46.32 | 131 |

**Additional Results on CIFAR-10:** This section expands on the CIFAR-10 results presented in Table 1 in Section 4.2 of the main manuscript by adding additional GDWS data points with different values of $\beta$. Table 3 shows that GDWS networks preserve $\mathcal{A}_{\text{nat}}$ and $\mathcal{A}_{\text{rob}}$ as both are within $\sim 1\%$ of their respective baselines. This further supports our claims in Section 4.2 that GDWS networks drastically improve the FPS while preserving robustness.

Table 3: Benefits of applying GDWS to standard pre-trained models on the CIFAR-10 dataset.

| Models | $\mathcal{A}_{\mathbf{nat}}$ [%] | $\mathcal{A}_{\mathbf{rob}}$ [%] | Size [MB] | FPS |
|---|---|---|---|---|
| ResNet-50 | 84.21 | 53.05 | 89.7 | 16 |
| + GDWS ($\beta = 0.001$) | 83.72 | 52.94 | 81.9 | 37 |
| + GDWS ($\beta = 0.005$) | 81.18 | 51.25 | 75.9 | 39 |
| WRN-28-4 | 84.00 | 51.80 | 22.3 | 17 |
| + GDWS ($\beta = 5 \times 10^{-6}$) | 83.64 | 51.62 | 19.9 | 64 |
| + GDWS ($\beta = 1 \times 10^{-5}$) | 83.27 | 51.70 | 18.9 | 65 |
| ResNet-18 | 82.41 | 51.55 | 42.6 | 28 |
| + GDWS ($\beta = 0.001$) | 82.17 | 51.30 | 33.5 | 89 |
| + GDWS ($\beta = 0.005$) | 81.17 | 50.98 | 29.1 | 104 |
| VGG-16 | 77.49 | 48.92 | 56.2 | 36 |
| + GDWS ($\beta = 0.1$) | 77.59 | 49.36 | 33.3 | 115 |
| + GDWS ($\beta = 0.25$) | 77.17 | 49.56 | 28.7 | 129 |

**New Results on SVHN:** Table 4 shows that applying GDWS to pre-trained networks on SVHN maintains the robustness while offering significant improvements in FPS, which mirrors the same observations made on CIFAR-10.

Table 4: Benefits of applying GDWS to standard pre-trained models on the SVHN dataset.

| Models | $\mathcal{A}_{\mathbf{nat}}$ [%] | $\mathcal{A}_{\mathbf{rob}}$ [%] | Size [MB] | FPS |
|---|---|---|---|---|
| WRN-28-4 | 90.71 | 52.27 | 22.3 | 17 |
| + GDWS ($\beta = 0.0001$) | 90.67 | 51.89 | 22.3 | 56 |
| + GDWS ($\beta = 0.0005$) | 90.60 | 51.11 | 22.1 | 64 |
| ResNet-18 | 88.63 | 55.57 | 42.6 | 28 |
| + GDWS ($\beta = 5 \times 10^{-5}$) | 87.87 | 55.88 | 39.9 | 80 |
| + GDWS ($\beta = 7.5 \times 10^{-5}$) | 87.37 | 55.66 | 39.3 | 89 |
| VGG-16 | 90.72 | 51.51 | 56.2 | 36 |
| + GDWS ($\beta = 0.1$) | 90.62 | 51.84 | 53.6 | 93 |
| + GDWS ($\beta = 5$) | 88.09 | 54.48 | 43.3 | 125 |

## 2.2 Additional Comparisons with HYDRA

In this section, we expand on the HYDRA [7] comparison in Section 4.2 by: 1) providing additional GDWS networks obtained with different values of $\beta$ presented in Table 5, 2) offering more insight

Table 5: Comparison between HYDRA [7] and GDWS using VGG-16 and WRN-28-4 on CIFAR-10.

| Models | $\mathcal{A}_{\textbf{nat}}$ [%] | $\mathcal{A}_{\textbf{rob}}$ [%] | Size [MB] | FPS |
|---|---|---|---|---|
| VGG-16 (AT from [7]) | 82.72 | 51.93 | 58.4 | 36 |
| + GDWS ($\beta = 0.1$) | 82.57 | 51.48 | 56.5 | 82 |
| + GDWS ($\beta = 0.5$) | 82.53 | 50.96 | 50.6 | 102 |
| + GDWS ($\beta = 1.2$) | 81.41 | 47.88 | 44.2 | 111 |
| VGG-16 ($p = 90\%$) | 80.54 | 49.44 | 5.9 | 36 |
| + GDWS ($\beta = 0.1$) | 80.47 | 49.52 | 31.5 | 93 |
| + GDWS ($\beta = 2$) | 78.52 | 47.26 | 26.9 | 101 |
| VGG-16 ($p = 95\%$) | 78.91 | 48.74 | 3.0 | 36 |
| + GDWS ($\beta = 0.1$) | 78.71 | 48.53 | 18.3 | 106 |
| + GDWS ($\beta = 0.5$) | 77.43 | 46.99 | 17.1 | 117 |
| VGG-16 ($p = 99\%$) | 73.16 | 41.74 | 0.6 | 41 |
| + GDWS ($\beta = 0.01$) | 72.88 | 41.79 | 3.0 | 130 |
| + GDWS ($\beta = 0.02$) | 72.75 | 41.56 | 2.9 | 136 |
| WRN-28-4 (AT from [7]) | 85.35 | 57.23 | 22.3 | 17 |
| + GDWS ($\beta = 0.01$) | 85.33 | 57.23 | 22.3 | 53 |
| + GDWS ($\beta = 0.5$) | 84.90 | 56.74 | 21.5 | 61 |
| + GDWS ($\beta = 1$) | 84.17 | 55.87 | 20.5 | 68 |
| WRN-28-4 ($p = 90\%$) | 83.69 | 55.20 | 2.3 | 17 |
| + GDWS ($\beta = 0.125$) | 83.38 | 54.79 | 11.9 | 59 |
| + GDWS ($\beta = 0.4$) | 81.21 | 52.01 | 11.4 | 65 |
| WRN-28-4 ($p = 95\%$) | 82.68 | 54.18 | 1.1 | 17 |
| + GDWS ($\beta = 0.005$) | 82.59 | 54.22 | 7.2 | 60 |
| + GDWS ($\beta = 0.1$) | 80.98 | 52.60 | 6.9 | 65 |
| WRN-28-4 ($p = 99\%$) | 75.62 | 47.21 | 0.2 | 28 |
| + GDWS ($\beta = 0.001$) | 75.46 | 47.30 | 1.3 | 66 |
| + GDWS ($\beta = 0.0025$) | 75.36 | 47.04 | 1.2 | 68 |

to why HYDRA pruned networks achieve limited FPS improvement compared to their un-pruned baselines, and 3) explaining why GDWS accelerates HYDRA pruned networks without any loss in robustness.

As seen in Section 4.2, Table 5 shows that HYDRA pruned models do not achieve significant improvements in FPS compared to their un-pruned baselines. The reason is that, despite having arbitrarily sparse weight matrices, the filter sparsity is actually quite low. That is the number of prunable channels in HYDRA pruned models is small, especially for pruning ratios less than 95%. To further demonstrate that effect, Figs 2a & 2b plot the per-layer filter sparsity of HYDRA pruned VGG-16 and WideResNet-28-4, respectively. These models are obtained from the publicly released CIFAR-10 HYDRA trained models available on GitHub. The plots indicate that only at *extreme* pruning ratios such as 99% does the filter sparsity in both networks appear to be significant, which translates to some improvement in FPS on the Jetson.

Table 5 also demonstrates that the application of GDWS to HYDRA-pruned networks provides significant improvement in FPS at iso-robustness when compared to the pruned baselines' numbers. Furthermore, the resultant GDWS networks are also sparse, which provide decent compression ratios. This synergy is due to the following observation: extremely sparse convolutional weight matrices have sub-matrices with low rank. This allows GDWS to transform standard sparse 2D convolutions into GDWS ones with no approximation error. The resultant GDWS convolutions are also sparse, which explains the improved compression ratios when compared to applying GDWS to un-pruned networks. To further understand this synergy, consider the following toy example: A standard $(3, 2, 4)$

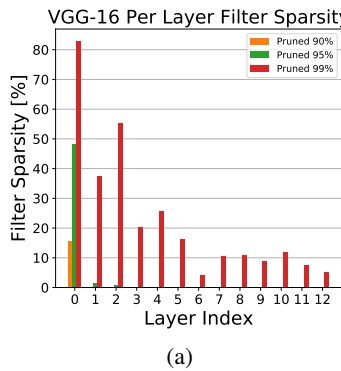
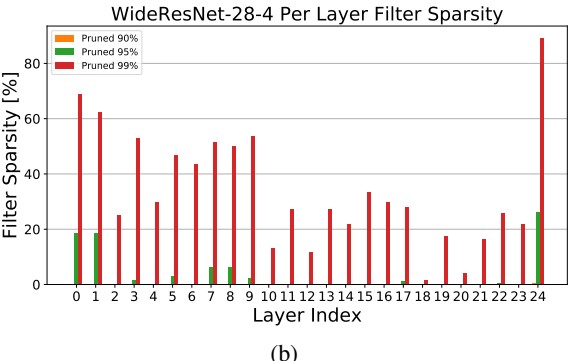

(a)                                                  (b)

Figure 2: Per-layer filter sparsity of HYDRA pruned VGG-16 and WideResNet-28-4 models on CIFAR-10.

2D convolution with pruned weight matrix $\mathbf{W} = [\mathbf{W}_1 | \mathbf{W}_2 | \mathbf{W}_3]$:

$$\mathbf{W} = \begin{bmatrix} w_1 & 0 & 0 & 0 & 0 & 0 & 0 & 0 & 0 & 0 & 0 & 0 \\ 0 & 0 & 0 & 0 & w_2 & 0 & 0 & 0 & 0 & 0 & 0 & 0 \\ 0 & 0 & 0 & 0 & 0 & 0 & 0 & 0 & w_3 & 0 & 0 & 0 \\ 0 & 0 & w_4 & 0 & 0 & 0 & 0 & 0 & 0 & 0 & 0 & 0 \end{bmatrix} \tag{1}$$

where:

$$\mathbf{W}_1 = \begin{bmatrix} w_1 & 0 & 0 & 0 \\ 0 & 0 & 0 & 0 \\ 0 & 0 & 0 & 0 \\ 0 & 0 & w_4 & 0 \end{bmatrix} \quad \mathbf{W}_2 = \begin{bmatrix} 0 & 0 & 0 & 0 \\ w_2 & 0 & 0 & 0 \\ 0 & 0 & 0 & 0 \\ 0 & 0 & 0 & 0 \end{bmatrix} \quad \mathbf{W}_3 = \begin{bmatrix} 0 & 0 & 0 & 0 \\ 0 & 0 & 0 & 0 \\ w_3 & 0 & 0 & 0 \\ 0 & 0 & 0 & 0 \end{bmatrix} \tag{2}$$

and $w_i \neq 0 \ \forall i \in \{1, 2, 3, 4\}$. Clearly, the weight matrx $\mathbf{W}$ does not have an all zero row, which implies that the filter sparsity is zero, despite having a high sparsity rate of $100 \times \frac{48-4}{48} = 91.\overline{6}\%$. However, we have that each sub-matrix $\mathbf{W}_c$ has low rank. Specifically: $\text{rank}(\mathbf{W}_1) = 2$ and $\text{rank}(\mathbf{W}_2) = \text{rank}(\mathbf{W}_3) = 1$, and computing the SVDs of each sub-matrix results in:

$$\begin{aligned} \mathbf{W}_1 &= \begin{bmatrix} w_1 \\ 0 \\ 0 \\ 0 \end{bmatrix} \begin{bmatrix} 1 & 0 & 0 & 0 \end{bmatrix} + \begin{bmatrix} 0 \\ 0 \\ 0 \\ w_4 \end{bmatrix} \begin{bmatrix} 0 & 0 & 1 & 0 \end{bmatrix} \\ \mathbf{W}_2 &= \begin{bmatrix} 0 \\ w_2 \\ 0 \\ 0 \end{bmatrix} \begin{bmatrix} 1 & 0 & 0 & 0 \end{bmatrix} \quad \mathbf{W}_3 = \begin{bmatrix} 0 \\ 0 \\ w_3 \\ 0 \end{bmatrix} \begin{bmatrix} 1 & 0 & 0 & 0 \end{bmatrix} \end{aligned} \tag{3}$$

Thus, from Lemma 1, we can construct a $(3, 2, \mathbf{g}, 4)$ GDWS convolution, where $\mathbf{g} = [2, 1, 1]^{\mathrm{T}}$, without any approximation error. Decomposing into a GDW matrix and a PW matrix results in:

$$\mathbf{W}_{\mathrm{PW}} = \begin{bmatrix} w_1 & 0 & 0 & 0 \\ 0 & 0 & w_2 & 0 \\ 0 & 0 & 0 & w_3 \\ 0 & w_4 & 0 & 0 \end{bmatrix} \quad \mathbf{W}_{\mathrm{GDW}} = \begin{bmatrix} 1 & 0 & 0 & 0 & 0 & 0 & 0 & 0 & 0 & 0 & 0 & 0 \\ 0 & 0 & 1 & 0 & 0 & 0 & 0 & 0 & 0 & 0 & 0 & 0 \\ 0 & 0 & 0 & 0 & 1 & 0 & 0 & 0 & 0 & 0 & 0 & 0 \\ 0 & 0 & 0 & 0 & 0 & 0 & 0 & 1 & 0 & 0 & 0 & 0 \end{bmatrix} \tag{4}$$

which are also sparse matrices. The total reduction in MACs is $\frac{4 \times 4 + 4 \times 4}{48} = 1.5\times$, and the total number of non-zero weights is $4 + 4 = 8$. This shows how extremely sparse standard 2D convolutions can be transformed into sparse GDWS convolutions with no approximation error while achieving improvements in complexity, which further justifies the synergy between GDWS and HYDRA pruned models.

# 3 Proofs

In this section we provide proofs for Theorems 1 and 2 stated in Section 3 of the main manuscript. We first state the following result due to Eckart and Young [1] on low-rank matrix approximations:

**Lemma** (Eckart-Young). *Let $\mathbf{A} \in \mathbb{R}^{m \times n}$ be an arbitrary rank $r$ matrix with the singular value decomposition $\mathbf{A} = \mathbf{U}\mathbf{\Sigma}\mathbf{V}^{\mathrm{T}} = \sum_{i=1}^{r} \sigma_i \mathbf{u}_i \mathbf{v}_i^{\mathrm{T}}$, such that $\sigma_1 \geq \sigma_2 \geq ... \geq \sigma_r > 0$. Define for all $p \in \{1, 2, ..., r-1\}^1$ the matrix $\hat{\mathbf{A}}_p$:*

$$\hat{\mathbf{A}}_p = \sum_{i=1}^{p} \sigma_i \mathbf{u}_i \mathbf{v}_i^{\mathrm{T}} \tag{5}$$

*Then $\hat{\mathbf{A}}_p$ is the optimal rank $p$ approximation in both the following senses:*

$$\min_{\mathbf{B}, \mathrm{rank}(\mathbf{B}) \leq p} ||\mathbf{A} - \mathbf{B}||_2 = \sigma_{p+1} \tag{6}$$

$$\min_{\mathbf{B}, \mathrm{rank}(\mathbf{B}) \leq p} ||\mathbf{A} - \mathbf{B}||_{\mathrm{F}} = \sqrt{\sum_{i=p+1}^{r} \sigma_i^2} \tag{7}$$

The Eckart-Young Lemma states that the truncated SVD can be used to compute the optimal rank $p$ approximation of any matrix in both the Frobenius norm and spectral norm sense. It also provides a closed form expression for the approximation error in terms of the singular values of the original matrix $\mathbf{A}$.

## 3.1 Proof of Lemma 1

**Lemma.** *The weight matrix $\mathbf{W} \in \mathbb{R}^{M \times CK^2}$ of a $(C, K, \mathbf{g}, M)$ GDWS convolution with $M > K^2$ can be expressed as the concatenation $[\mathbf{W}_1|...|\mathbf{W}_C]$ of $C$ sub-matrices $\mathbf{W}_c \in \mathbb{R}^{M \times K^2}$ such that $\mathrm{rank}(\mathbf{W}_c) \leq \min(g_c, K^2) \ \forall c \in [C]$.*

*Proof.* From Property 1, $\mathbf{W}$ is decomposed as:

$$\mathbf{W} = [\mathbf{W}_1|...|\mathbf{W}_C] = \mathbf{W}_{\mathrm{P}}\mathbf{W}_{\mathrm{D}} = [\mathbf{u}_1|...|\mathbf{u}_G] \times [\mathbf{W}_{\mathrm{D},1}|...|\mathbf{W}_{\mathrm{D},C}] \tag{8}$$

where $\mathbf{u}_i \in \mathbb{R}^M$ are the column vectors of $\mathbf{W}_{\mathrm{P}}$ and $\mathbf{W}_{\mathrm{D},c}^{\mathrm{T}} \in \mathbb{R}^{K^2 \times G} = [\mathbf{v}_{1,c}|...|\mathbf{v}_{G,c}]$ has column vectors $\mathbf{v}_{i,c} \in \mathbb{R}^{K^2}$. From (8), the sub-matrix $\mathbf{W}_c \in \mathbb{R}^{M \times K^2}$ is given by:

$$\mathbf{W}_c = \mathbf{W}_{\mathrm{P}}\mathbf{W}_{\mathrm{D},c} = \sum_{i=1}^{G} \mathbf{u}_i \mathbf{v}_{i,c}^{\mathrm{T}} = \sum_{i=1+h_c}^{h_c+g_c} \mathbf{u}_i \mathbf{v}_{i,c}^{\mathrm{T}} \tag{9}$$

with $h_c = \sum_{k=1}^{c-1} g_k$ where we employ Property 2 to obtain the rightmost equality. Therefore, $\mathbf{W}_c$ is a sum of $g_c$ rank 1 matrices $\mathbf{u}_i \mathbf{v}_{i,c}^{\mathrm{T}} \in \mathbb{R}^{M \times K^2}$ which implies $\mathrm{rank}(\mathbf{W}_c) \leq \min(g_c, K^2)$. This concludes the proof. □

## 3.2 Proof of Theorem 1

**Definition**: The *weighted approximation error* between two matrices $\mathbf{W} = [\mathbf{W}_1|...|\mathbf{W}_C]$ and $\mathbf{Q} = [\mathbf{Q}_1|...|\mathbf{Q}_C]$ is

$$e(\mathbf{W}, \mathbf{Q}, \boldsymbol{\alpha}) = \sqrt{\sum_{c=1}^{C} \alpha_c ||\mathbf{W}_c - \mathbf{Q}_c||_{\mathrm{F}}^2} \tag{10}$$

where $\boldsymbol{\alpha} \in \mathbb{R}_+^C$ and all sub-matrices $\mathbf{W}_c$ and $\mathbf{Q}_c$ have the same size.

We first prove the following Lemma:

---

[1] when $p = r$, the summation in (7) becomes undefined, but the error is zero.

**Lemma 2.** *Given any $(C, K, M)$ standard 2D convolution with weight matrix $\mathbf{W} = [\mathbf{W}_1|...|\mathbf{W}_C] \in \mathbb{R}^{M \times CK^2}$, $\mathbf{W}_c \in \mathbb{R}^{M \times K^2}$ and $K^2 < M$, the $(C, K, \mathbf{g}, M)$ GDWS approximation with weight matrix $\hat{\mathbf{W}}$ and fixed channel distribution vector $\mathbf{g}$ that minimizes the weighted approximation error $e(\mathbf{W}, \hat{\mathbf{W}}, \boldsymbol{\alpha})$ with $\boldsymbol{\alpha} \in \mathbb{R}_+^C$ is obtained via the concatenation $\hat{\mathbf{W}} = [\hat{\mathbf{W}}_1|...|\hat{\mathbf{W}}_C]$, where $\hat{\mathbf{W}}_c$ is the optimal rank $g_c$ approximation of $\mathbf{W}_c$.*

*Proof.* Since $\mathbf{W}_c \in \mathbb{R}^{M \times K^2}$ with $K^2 < M$, we have $\text{rank}(\mathbf{W}_c) = r_c \leq K^2$. Let $\mathbf{Q} = [\mathbf{Q}_1|...|\mathbf{Q}_C] \in \mathbb{R}^{M \times CK^2}$ be the weight matrix of a $(C, K, \mathbf{g}, M)$ GDWS convolution. Then, from Lemma 1, we have that $\text{rank}(\mathbf{Q}_c) \leq \min(g_c, K^2)$. Without loss of generality, we will always assume $g_c \leq r_c \leq K^2$, since otherwise $g_c > r_c$ for some $c$ implies the optimal rank $g_c$ approximation of $\mathbf{W}_c$ is $\hat{\mathbf{W}}_c = \mathbf{W}_c$ resulting in at most $r_c$ non-zero DW kernels in the $c^{\text{th}}$ channel and $g_c - r_c$ zero DW kernels.

Then, from the Eckart-Young Lemma, we obtain:

$$||\mathbf{W}_c - \mathbf{Q}_c||_{\text{F}} \geq \sqrt{\sum_{i=g_c+1}^{r_c} \sigma_{i,c}^2} = ||\mathbf{W}_c - \hat{\mathbf{W}}_c||_{\text{F}} \tag{11}$$

where $\sigma_{1,c} \geq \sigma_{2,c} \geq ... \geq \sigma_{r_c,c} > 0$ are the singular values of $\mathbf{W}_c \ \forall c \in [C]$ and $\hat{\mathbf{W}}_c = \sum_{i=1}^{g_c} \sigma_{i,c} \mathbf{u}_{i,c} \mathbf{v}_{i,c}^{\text{T}}$ is its rank $g_c$ truncated SVD. The equality holds if and only if $\mathbf{Q}_c = \hat{\mathbf{W}}_c$.

For a fixed $\mathbf{g}$, we have:

$$e^2(\mathbf{W}, \mathbf{Q}, \boldsymbol{\alpha}) = \sum_{c=1}^{C} \alpha_c ||\mathbf{W}_c - \mathbf{Q}_c||_{\text{F}}^2 \geq \sum_{c=1}^{C} \alpha_c \sum_{i=g_c+1}^{r_c} \sigma_{i,c}^2 = e^2(\mathbf{W}, \hat{\mathbf{W}}, \boldsymbol{\alpha}) \tag{12}$$

where $\hat{\mathbf{W}} = [\hat{\mathbf{W}}_1|...|\hat{\mathbf{W}}_C]$. This completes the proof since minimizing $e^2$ also minimizes $e$. $\square$

We now prove Theorem 1:

**Theorem.** *Given any $(C, K, M)$ standard 2D convolution with weight matrix $\mathbf{W}$, the $(C, K, \mathbf{g}, M)$ GDWS approximation with weight matrix $\hat{\mathbf{W}}$ that minimizes the weighted approximation error $e(\mathbf{W}, \hat{\mathbf{W}}, \boldsymbol{\alpha})$ with $\boldsymbol{\alpha} \in \mathbb{R}_+^C$ subject to $\sum g_c = G \leq \gamma$ (for some $\gamma \in \mathbb{Z}_+$), can be obtained in polynomial time via Algorithm 1.*

*Proof.* We want to show that:
$$\hat{\mathbf{W}} = \underset{\mathbf{Q}: \ G \leq \gamma}{\arg \min} e(\mathbf{W}, \mathbf{Q}, \boldsymbol{\alpha}) \tag{13}$$

can be solved optimally for any $\boldsymbol{\alpha} \in \mathbb{R}_+^C$. We show this using an induction on the constraint $\gamma$ via a constructive proof, which provides the basis for Algorithm 1. Essentially, we show that solving (13) with constraint $\gamma + 1$ can be obtained from the solution of (13) with constraint $\gamma$ via a 1D search over the channels $C$, and establish the base case for when $\gamma = 1$.

Without any loss of generality, we will assume that $\alpha_c > 0 \ \forall c \in [C]$. The reason for this is that if $\alpha_c = 0$ for a particular $c$, then we can set $\hat{\mathbf{W}}_c = \mathbf{0}$ in the optimal solution and have $g_c = 0$ which minimizes the complexity and does not contribute to the error expression. Similar to before, let $\mathbf{W} = [\mathbf{W}_1|...|\mathbf{W}_C]$ be the concatenation of $C$ sub-matrices. We have $\text{rank}(\mathbf{W}_c) = r_c \leq K^2$. Let the SVD of each sub-matrix be:

$$\mathbf{W}_c = \mathbf{U}_c \mathbf{\Sigma}_c \mathbf{V}_c^{\text{T}} = \sum_{i=1}^{r_c} \sigma_{i,c} \mathbf{u}_{i,c} \mathbf{v}_{i,c}^{\text{T}} \tag{14}$$

where $\sigma_{1,c} \geq \sigma_{2,c} \geq ... \geq \sigma_{r_c,c} > 0$ are the singular values of $\mathbf{W}_c \ \forall c \in [C]$.

Assume that $\hat{\mathbf{W}}^{(\gamma)}$ is the optimal solution to (13) with constraint $\gamma$, that is $\hat{\mathbf{W}}^{(\gamma)}$ corresponds to a $(C, K, \mathbf{g}^{(\gamma)}, M)$ GDWS convolution with channel distribution vector $\mathbf{g}^{(\gamma)} \in \mathbb{Z}_+^C$ such that $G^{(\gamma)} = $

$\sum g_c^{(\gamma)} \leq \gamma$. From Lemma 2, we have $\hat{\mathbf{W}}^{(\gamma)} = [\hat{\mathbf{W}}_1^{(\gamma)}|...|\hat{\mathbf{W}}_C^{(\gamma)}]$ such that $\text{rank}(\hat{\mathbf{W}}_c^{(\gamma)}) \leq g_c^{(\gamma)} \leq r_c$, with optimal weighted approximation error:

$$e^2(\mathbf{W}, \hat{\mathbf{W}}^{(\gamma)}, \boldsymbol{\alpha}) = \sum_{c=1}^{C} \alpha_c \sum_{i=g_c^{(\gamma)}+1}^{r_c} \sigma_{i,c}^2 \tag{15}$$

Then, solving (13), with constraint $G \leq \gamma + 1$ will result in a $(C, K, \mathbf{g}^{(\gamma+1)}, M)$ GDWS convolution such that the channel distribution vector $\mathbf{g}^{(\gamma+1)}$ will differ from $\mathbf{g}^{(\gamma)}$ in at most one position $c' \in [C]$, such that $g_{c'}^{(\gamma+1)} = g_{c'}^{(\gamma)} + 1$. The reason for this is that: 1) $G^{(\gamma+1)} \geq G^{(\gamma)}$, otherwise the optimal solution for the $\gamma$ constraint could be improved; and 2) the integer constraints on both $G^{(\gamma+1)}$ and $G^{(\gamma)}$ imply that their difference can be at most 1, and hence the corresponding vectors will be identical up to one position. Thus, the optimal approximation error with constraint $\gamma + 1$ can be computed from $e^2(\mathbf{W}, \hat{\mathbf{W}}^{(\gamma)}, \boldsymbol{\alpha})$:

$$e^2(\mathbf{W}, \hat{\mathbf{W}}^{(\gamma+1)}, \boldsymbol{\alpha}) = e^2(\mathbf{W}, \hat{\mathbf{W}}^{(\gamma)}, \boldsymbol{\alpha}) - \max_{c \in [C]: g_c^{(\gamma)} < r_c} \alpha_c \sigma_{g_c^{(\gamma)}+1,c}^2 \tag{16}$$

where the maximization is taken over all channels $c$ that are not saturated (that is $g_c^{(\gamma)} + 1 \leq r_c$ is valid). If no such channels exist, then the approximation error is saturated, and there is no point in increasing complexity further, which implies $\mathbf{g}^{(\gamma+1)} = \mathbf{g}^{(\gamma)}$. Therefore, we can construct the optimal channel distribution vector $\mathbf{g}^{(\gamma+1)}$ from $\mathbf{g}^{(\gamma)}$ as previously mentioned, and then use Lemma 2 to find $\hat{\mathbf{W}}^{(\gamma+1)}$.

Lastly, we show how to solve (13) for the smallest constraint $\gamma = 1$, which establishes the base case, and thus concludes the proof. Notice that, if $\gamma = 1$, then $G = 1$, and $\mathbf{g}$ reduces to the basis vector $\mathbf{e}_c$ (vector of all zeros except for one position $c$ such that $e_c = 1$). Thus the optimal GDWS approximation with $G = 1$ can be solved by simply searching for the channel $c$ that maximizes $\alpha_c \sigma_{1,c}^2$, and then use Lemma 2 to find $\hat{\mathbf{W}}^{(1)}$. $\qquad\square$

### 3.3 Proof of Theorem 2

**Theorem.** *Given any $(C, K, M)$ standard 2D convolution with weight matrix $\mathbf{W}$, the $(C, K, \mathbf{g}, M)$ GDWS approximation with weight matrix $\hat{\mathbf{W}}$ that minimizes the complexity captured by $G = \sum g_c$ subject to $e(\mathbf{W}, \hat{\mathbf{W}}, \boldsymbol{\alpha}) \leq \beta$ (for some $\beta \geq 0$ and $\boldsymbol{\alpha} \in \mathbb{R}_+^C$), can be constructed in polynomial time via Algorithm 2.*

*Proof.* We want to show that:

$$\hat{\mathbf{W}} = \underset{\mathbf{Q}: e(\mathbf{W}, \mathbf{Q}, \boldsymbol{\alpha}) \leq \beta}{\arg\min} \sum_{c=1}^{C} g_c \tag{17}$$

can be solved for any weight error vector $\boldsymbol{\alpha} \in \mathbb{R}_+^C$ in polynomial time. We show this by first applying a re-formulation of both the objective and the constraint as a function of a single binary vector. Using this new formulation, we show that solving (17) reduces to a greedy approach, captured in Algorithm 2, consisting of a simple 1D search over sorted quantities.

Similar to before, let $\mathbf{W} = [\mathbf{W}_1|...|\mathbf{W}_C]$ be the concatenation of $C$ sub-matrices. We have $\text{rank}(\mathbf{W}_c) = r_c \leq K^2$. Let the SVD of each sub-matrix be:

$$\mathbf{W}_c = \mathbf{U}_c \boldsymbol{\Sigma}_c \mathbf{V}_c^\mathsf{T} = \sum_{i=1}^{r_c} \sigma_{i,c} \mathbf{u}_{i,c} \mathbf{v}_{i,c}^\mathsf{T} \tag{18}$$

where $\sigma_{1,c} \geq \sigma_{2,c} \geq ... \geq \sigma_{r_c,c} > 0$ are the singular values of $\mathbf{W}_c \; \forall c \in [C]$. Furthermore, without loss of generality we will assume that $\alpha_c > 0 \; \forall c \in [C]$. For a fixed channel distribution vector $\mathbf{g}$, weight error vector $\boldsymbol{\alpha}$ and convolution matrix $\mathbf{W}$, Lemma 2 states that the optimal GDWS approximation error can be computed via:

$$e^2(\mathbf{g}) = e^2(\mathbf{W}, \hat{\mathbf{W}}, \boldsymbol{\alpha}) = \sum_{c=1}^{C} \alpha_c \sum_{i=g_c+1}^{r_c} \sigma_{i,c}^2 \tag{19}$$

Therefore, for any $\beta \geq 0$, there always exists a GDWS convolution satisfying $e(\mathbf{g}) \leq \beta$. A simple choice of $g_c = r_c \ \forall c \in [C]$ will result in $e(\mathbf{r}) = 0 \leq \beta$, where $\mathbf{r} \in \mathbb{Z}_+^C$ is the vector of sub-matrix ranks $r_c$'s. The goal is to find the least complex GDWS convolution, satisfying the constraint.

Let $\mathcal{A}$ be an ordered set of all $R = \sum r_c$ quantities $\alpha_c \sigma_{i,c}^2$. Define an indexing $k \in [R]$ on $\mathcal{A}$ where $a_k \in \mathcal{A}$ corresponds to a unique pair $(i, c)$ such that $a_k = \alpha_c \sigma_{i,c}^2$ and $a_1 \geq a_2 \geq ... \geq a_R > 0$. By doing so, we can re-write the error expression (19):

$$e^2(\mathbf{g}) = \sum_{c=1}^{C} \alpha_c \sum_{i=g_c+1}^{r_c} \sigma_{i,c}^2 = \sum_{k=1}^{R} a_k t_k \tag{20}$$

where $t_k \in \{0, 1\}$ are binary variables indicating whether the corresponding pair $(i, c)$ exists in the original sum in (19). This change of variables facilitates the optimization problem in (17), since the binary vector $\mathbf{t} \in \{0, 1\}^R$ can be used to enumerate all possible GDWS approximations with a simple expression of the optimal error in (20). Another useful thing about this re-formulation is the following property:

$$\sum_{c=1}^{C} g_c = \sum_{k=1}^{R} \bar{t}_k = G \tag{21}$$

where $\bar{t}_k = |1 - t_k|$ is the flipped binary variable. Using the fact that the $\{a_k\}$'s are sorted in descending order, let $j \in [R]$ be the smallest index such that:

$$\sum_{k=j+1}^{R} a_k \leq \beta^2 \tag{22}$$

Then setting $t_k = 1 \ \forall k > j$ and $t_k = 0$ otherwise, will result in the least complex (least sum $\sum \bar{t}_k$) GDWS approximation satisfying the error constraint $e \leq \beta$. Finding the index $j$ can be done via a simple 1D search, by starting with $j = R$ (corresponding to the zero error case), and keep decrementing $j$ until the error condition is no longer satisfied. After finding the optimal vector $\mathbf{t}$, the corresponding unique channel distribution vector can be constructed via the index mapping:

$$g_c = \sum_{k \in \mathcal{K}_c} \bar{t}_k \tag{23}$$

where $\mathcal{K}_c \subset [R]$ is the set of indices $k$ such that the corresponding index pair $(i, c')$ satisfies $c' = c$. Finally, given the channel distribution vector $\mathbf{g}$, we can use Lemma 2 to construct $\hat{\mathbf{W}}$. The greedy algorithm presented in Algorithm 2 computes $\mathbf{g}$ via this approach, but without dealing with the auxiliary indexing and reformulation. $\qquad \square$

# 4 Rationale for the Weight Error Vector Expression

In this section, we provide a detailed explanation for our choice of $\alpha_l$ in (8) (from main manuscript). The work of [5] presents theoretical bounds on the accuracy of neural networks, in the presence of quantization noise due to quantizing both weights and activations, to determine the minimum precision required to maintain accuracy. A follow-up work [6] extends this bound to the per-layer precision case, allowing for better complexity-accuracy trade-offs. The bound in [5] in fact is much more general, and is not restricted to neural network quantization. Consider the following *scalar* additive perturbation model:

$$\hat{w} = w + \eta_w \tag{24}$$

where $\eta_w$ is assumed to be a zero-mean, symmetric and independently distributed scalar random variable with variance $s^2$. Then the work of [5, 6] shows that the probability $p_{\mathrm{m}}$ that the noisy network $\hat{f}$ paramerterized by $\hat{w}$ differs in its decision from $f$ can be upper bounded as follows:

$$p_{\mathrm{m}} \leq \sum_{l=1}^{L}\sum_{c=1}^{C_l} s_{c,l}^2 E_{c,l} \quad \text{where} \quad E_{c,l} = \mathbb{E}\left[\sum_{\substack{j=1 \\ j \neq n_x}}^{N} \frac{\sum_{w \in \mathcal{W}_c^{(l)}} \left|\frac{\partial \delta_{x,j}}{\partial w}\right|^2}{2\delta_{x,j}^2}\right] \tag{25}$$

where the following notation, inherited from Section 3.4 of the main manuscript, is used: Let $f : \mathbb{R}^D \to \mathbb{R}^N$ be a pre-trained CNN for an $N$-way classification problem with $L$ convolutional layers parameterizd by weight matrices $\mathbf{W}^{(l)} \in \mathbb{R}^{M_l \times C_l K_l^2}$. The CNN $f$ operates on a $D$-dimensional input vector $\mathbf{x}$ to produce a vector $\mathbf{z} = f(\mathbf{x})$ of soft outputs or logits. Denote by $n_x \in [N]$ the predicted class label associated with $\mathbf{x}$, and define $\delta_{x,j} = z_j - z_{n_x}$ to be the soft output differences $\forall j \in [N] \setminus \{n_x\}$. **In addition**, define $\mathcal{W}_c^{(l)}$ to be the set containing all the $C_l K_l^2$ scalar entries of sub-matrix $\mathbf{W}_c^{(l)}$ $\forall c \in [C_l]$ $\forall l \in [L]$, that is the cardinality of $\mathcal{W}_c^{(l)}$ is $C_l K_l^2$. Using this notation, $\mathcal{W}^{(l)} = \bigcup_c \mathcal{W}_c^{(l)}$ is essentially the set of all scalar parameters in the $l^{\mathrm{th}}$ convolutional layer, and $\mathcal{W} = \bigcup_l \mathcal{W}^{(l)}$ is the set of all scalar parameters of $f$ across all convolutional layers.

When approximating standard 2D convolutions with GDWS convolutions, we incur approximation errors that are captured at the sub-matrix level, and not at the entry level. Let $\mathbf{W}^{(l)} = [\mathbf{W}_1^{(l)}|\mathbf{W}_2^{(l)}|...|\mathbf{W}_C^{(l)}]$ be the weight matrix, and its corresponding sub-matrices, of the standard convolution for layer $l$. Define $r_c^{(l)} = \mathrm{rank}(\mathbf{W}_c^{(l)}) \leq K_l^2$. Similarly, let $\mathbf{Q}^{(l)} = [\mathbf{Q}_1^{(l)}|\mathbf{Q}_2^{(l)}|...|\mathbf{Q}_C^{(l)}]$ be the weight matrix, and its corresponding sub-matrices, of the GDWS convolution approximation for layer $l$. From Lemma 1 we know that $\mathrm{rank}(\mathbf{Q}_c^{(l)}) = g_c^{(l)}$. Then, based on the proofs in Appendix 3, the sub-matrix approximation error can be expressed as:

$$e_c^{(l)} = ||\mathbf{W}_c^{(l)} - \mathbf{Q}_c^{(l)}||_{\mathrm{F}} = ||\mathbf{R}_c^{(l)}||_{\mathrm{F}} = \sqrt{\sum_{i=g_c^{(l)}+1}^{r_c^{(l)}} \sigma_{i,c}^{(l)}{}^2} \tag{26}$$

where $\sigma_{1,c}^{(l)} \geq \sigma_{2,c}^{(l)} \geq ... \geq \sigma_{r_c^{(l)},c}^{(l)} > 0$ are the singular values of $\mathbf{W}_c^{(l)}$ $\forall c \in [C_l]$ $\forall l \in [L]$. Clearly, the setup in [5, 6] does not hold here. However, we circumvent this issue by assuming that for all entries $w \in \mathcal{W}_c^{(l)}$, the additive perturbation model in (24) holds where $\eta_w$ are additive, zero-mean, symmetric, independent random variables with variance:

$$\mathbb{E}\left[\eta_w{}^2\right] = \frac{||\mathbf{R}_c^{(l)}||_{\mathrm{F}}^2}{M_l K_l^2} \quad \forall c \in [C_l] \, \forall l \in [L] \tag{27}$$

While this assumption does not hold, it allows us to use the upper bound in (25) to provide a **heuristic** in our setup:

$$
\begin{aligned}
p_{\mathrm{m}} &\leq \sum_{l=1}^{L} \sum_{c=1}^{C_l} s_{c,l}^2 E_{c,l} \\
&= \sum_{l=1}^{L} \sum_{c=1}^{C_l} \frac{||\mathbf{R}_c^{(l)}||_{\mathrm{F}}^2}{M_l K_l^2} \; \mathbb{E} \left[ \sum_{\substack{j=1 \\ j \neq n_x}}^{N} \frac{\sum_{w \in \mathcal{W}_c^{(l)}} \left| \frac{\partial \delta_{x,j}}{\partial w} \right|^2}{2 \delta_{x,j}^2} \right] \\
&= \sum_{l=1}^{L} \sum_{c=1}^{C_l} \frac{||\mathbf{R}_c^{(l)}||_{\mathrm{F}}^2}{M_l K_l^2} \; \mathbb{E} \left[ \sum_{\substack{j=1 \\ j \neq n_x}}^{N} \frac{||\mathbf{D}_{x,j}^{(c,l)}||_{\mathrm{F}}^2}{2 \delta_{x,j}^2} \right] \qquad (28) \\
&= \sum_{l=1}^{L} \sum_{c=1}^{C_l} \alpha_{c,l} ||\mathbf{R}_c^{(l)}||_{\mathrm{F}}^2 \\
&= \sum_{l=1}^{L} e(\mathbf{W}^{(l)}, \mathbf{Q}^{(l)}, \boldsymbol{\alpha}_l)^2
\end{aligned}
$$

where $\alpha_{c,l}$ is the same as before, with the definition $\mathbf{D}_{x,j}^{(c,l)} \in \mathbb{R}^{M_l \times K_l^2}$ being the derivative of $\delta_{x,j}$ w.r.t. the sub-matrix $\mathbf{W}_c^{(l)}$. Thus, the upper bound on $p_{\mathrm{m}}$ results in a sum of $L$ terms, where each term is the GDWS approximation error. Following [6], we use noise gain equalization to minimize this sum. That is we make sure all the terms are of comparable magnitude by upper-bounding them with the same $\beta$ when using Algorithm 2.