# OpenReview forum: "Generalized Depthwise-Separable Convolutions for Adversarially Robust and Efficient Neural Networks"
_NeurIPS.cc/2021/Conference — NeurIPS 2021 Spotlight_

### Official Review · Reviewer_PpQM · 2021-07-13

**Rating:** 3
**Confidence:** 5

**Summary:**

The authors propose Generalized Depthwise-Separable (GDWS) convolutions for processing images for the purpose of getting high throughput in efficiency and robustness. This is carried out by introducing channel distribution vectors in the standard 2D CNNs. The efficiency is managed by means of the singular value decomposition of the matrix in the convolutional operations after vectorizations. Experiments show that the method provides similar results for some benchmark data.

**Limitations And Societal Impact:**

The authors admit the limitations of their work in not achieving high compression ratios and being unable to be applied to CNNs utilizing depthwise-separable convolutions. This is described in the session of discussion.

**Main Review:**

The problem of getting efficiency and robustness is important for applying 2-D CNNs to dealing with images. The idea of channel distribution vectors to get efficiency is interesting. However, this idea has not been fully explored in the paper. The reviewer has the following concerns:

1. Activation. In the proposed method, only the output layer involves an activation function. The GDWS does not involve any nonlinear activation functions in the hidden layers. So the role of depth and the effect of deep learning cannot be seen.

2. Vectorization. The technique of vectorization is often used for dealing with image data. But it breaks the structures of images and hidden layers, and the role of convolutions is not reflected in GDWS.

3. The experiment results do not demonstrate significant improvements of GDWS in improving efficiency and robustness of the standard CNNs.



**Time Spent Reviewing:**

4

---

> ### Author Response · Authors · 2021-08-10
> **Reply to Reviewer PpQM**
>
> Thank you for your review. First, we would like to clarify a few misunderstandings, and we apologize if the manuscript does not clearly explain the following:
>
> In this work, we propose GDWS, a generalized form of the popular DWS convolution where more than one DW kernel is allowed per input channel in the GDW layer with a subsequent PW layer operating on the intermediate feature map. In doing so, we show that any pre-trained robust network can be mapped to hardware with increased throughput (FPS) and minimal loss in robustness. A major advantage of our method is that it does not require any training unlike existing methods.
>
> With that being said, we hope that we address your concerns below:
>
> **“Activation. In the proposed method, only the output layer involves an activation function. The GDWS does not involve any nonlinear activation functions in the hidden layers. So the role of depth and the effect of deep learning cannot be seen.”:**
>
> We are not exactly sure what this comment means. If you mean that GDWS does not have a nonlinearity between the GDW and PW layers then it is because their combination is meant to approximate a standard 2D convolution. All the nonlinearities after the standard convolutions are preserved, same as with any other layer in the network. The depth and effect of deep learning is automatically included in the performance of the pre-trained network.
>
> **“Vectorization. The technique of vectorization is often used for dealing with image data. But it breaks the structures of images and hidden layers, and the role of convolutions is not reflected in GDWS.”:**
>
> We are not exactly sure what this comment means. If you are referring to the vectorization of a 2D convolution kernel, then this reformulation is there to simplify notation and the subsequent analysis. The implementation of GDWS in hardware does not require vectorization, if that is your concern.
>
> **“The experiment results do not demonstrate significant improvements of GDWS in improving efficiency and robustness of the standard CNNs.”:**
>
> We beg to differ. In fact, Figure 1 itself demonstrates the improvement in robustness and FPS of GDWS when compared to existing robust complexity reduction techniques. Furthermore, we present a comprehensive set of experiments, across three datasets (CIFAR-10, SVHN, and ImageNet), multiple network architectures (VGG-16, ResNet-18, ResNet-50, WideResNet-28-4), and different adversarial perturbation models ($\ell_\infty$ and union of ($\ell_\infty$, $\ell_2$, $\ell_1$)) all showcasing how GDWS networks are able to significantly boost the FPS of standard pre-trained networks while maintaining accuracy and robustness.
>
> Finally, we hope we have addressed some if not all of your concerns, and hope you would give us a second chance when updating your review.

---

> > ### Comment · Reviewer_PpQM · 2021-08-26
> > **I read the response and would like to keep my rating.**
> >
> > My rating is kept.

---

> > > ### Author Response · Authors · 2021-08-30
> > > **Response to Reviewer PpQM**
> > >
> > > We are disappointed that our rebuttal did not alter your opinion. If we misunderstood your questions/concerns, please do clarify and we will be happy to respond promptly.

---

### Official Review · Reviewer_c4Yo · 2021-07-14

**Rating:** 6
**Confidence:** 3

**Summary:**

The paper proposes a generalization of depthwise-separable convolutions (called GDWS) where each channel can have multiple depthwise filters (instead of just one) that are subsequently combined using a pointwise convolution. Two algorithms are proposed to minimize the error/complexity of mapping general convolutions to GDWS as a post-processing step after training. Results show small degradation on natural and robust accuracy while increasing the FPS substantially.

**Limitations And Societal Impact:**

Sufficiently addressed (limitations: see above)

**Main Review:**

The paper is well written (easy to follow) and considers an important topic. The idea of generalizing depthwise-separable convolutions as proposed in the paper seems novel to me and seems to substantially increase FPS while only having a small negative impact on accuracy.

The paper provides an analysis showing that a general convolution can be written as a GDWS convolution where each channel has K^2 depthwise filters. This is a good justification for approximating general convolutions using fewer depthwise filters, but using more than a single filter as in conventional depthwise-separable convolutions.

I wonder why the proposed method is suitable for adversarial robustness in particular. Is it because the method maintains the adversarial robustness of a pre-trained neural network due to the good approximation quality? Or is there something special to GDWS that is particularly well suited to achieve adversarial robustness? At least I have not figured this out after reading the paper and some insights are required. It does not appear that the proposed method is specifically tailored to improving adversarial robustness, but that this is rather a byproduct that is empirically evaluated.

As mentioned in Section 5, I would have liked to see some experiments where GDWS is used to train from scratch or, at least, to apply some fine-tuning after applying Algorithm 3 to regain some of the lost accuracy. This is a standard practice of methods that approximate building blocks of pre-trained methods using cheaper building blocks (e.g., pruning). Did the authors try this? If yes: Does it help? Does it harm the adversarial robustness?

How much do the results regarding throughput depend on the hardware and the implementation? Since the increased throughput seems to be a major selling point of the proposed method, this also needs to be evaluated on some other non-GPU/TPU hardware without massive parallelism. Furthermore, I have the feeling that the comparison with unstructured pruning is unfair. Unstructured pruning can benefit a lot if suitable sparse tensor structures are used. However, in the proposed setting it seems that only those filters are removed that only contain zero entries, and the rest is treated as a dense tensor. I believe that a network with 95% sparsity can have way more throughput than one without any sparsity, but the reported results have the same FPS.

I would have liked to see some experiments that show how the individual layers are affected by the compression, e.g., how does the average g_c look like for individual layers? For instance, is there some difference between input/output layers and intermediate layers? From the reported FPS it is difficult to tell where the increased throughput stems from. Maybe you can also provide some intuition why the FPS is affected more severely than the size. It would also be interesting to see how the accuracy behaves for different g_c / different approximation errors.

Just to be sure: Is the "natural accuracy" the accuracy on the standard test dataset (without adversarial perturbations)? In this case, can you comment on the rather weak natural accuracies on the given datasets (e.g., approx 83% is rather weak on Cifar-10).

minor:
- Figure 1a: What are the numbers in brackets?

**Time Spent Reviewing:**

5

---

> ### Author Response · Authors · 2021-08-10
> **Reply to Reviewer c4Yo**
>
>
> Thank you for the positive review. We hope that we address your comments below:
>
> **“I wonder why the proposed method is suitable for adversarial robustness in particular. Is it because the method maintains the adversarial robustness of a pre-trained neural network due to the good approximation quality? ”:**
>
> That is correct. GDWS convolutions are more efficient approximations of standard 2D convolutions. Therefore, a GDWS network, with small enough approximation error, inherits all the accuracy/robustness properties from the baseline network with significantly improved FPS.
>
> **“is there something special to GDWS that is particularly well suited to achieve adversarial robustness?“:**
>
> There is nothing special to GDWS w.r.t. adversarial robustness other than its post-training attribute that makes it suitable to design high FPS robust networks. Unlike existing methods, GDWS does not require any training beyond standard adversarial training which is already very expensive as is.
>
> **“I would have liked to see some experiments where GDWS is used to train from scratch”:**
>
> Thank you for bringing this up. Training GDWS networks from scratch is a distinct possibility. However, this is beyond the scope of this paper because there are a few challenges that need to be addressed, e.g., in choosing the per-layer channel distribution vectors ($\mathbf{g}$). This is definitely an excellent future research direction.
>
> **“or, at least, to apply some fine-tuning … did the authors try this? If yes: Does it help? Does it harm the adversarial robustness?”:**
>
> Excellent suggestion. We did try fine-tuning (see data below) but wanted to preserve the “no training cost” attribute of GDWS in this paper. Even without fine-tuning we are able to boost the FPS of pre-trained models. In the table below, we fine-tune after GDWS. As expected, fine-tuning boosts the efficacy of GDWS even further, as seen in the table below:
>
>
> |          Model         | $A_{\text{nat}}$ (after fine-tuning) | $A_{\text{rob}}$ (after fine-tuning) | Size [MB] | FPS |
> |:----------------------:|:------------------------------------------------:|:------------------------------------------------:|:---------:|:---:|
> |         VGG-16         |                    77.49 (NA)                    |                       48.92 (NA)                 |    56.2   |  36 |
> | + GDWS ($\beta = 2$) |                   72.05 (77.15)                  |                   45.35 (47.87)                  |    19.1   | 140 |
> | + GDWS ($\beta = 5$) |                   63.21 (76.76)                  |                   37.78 (47.92)                  |    16.3   | 143 |
>
>
> We can include these results in the supplementary material.
>
>
> **“How much do the results regarding throughput depend on the hardware and the implementation? … this also needs to be evaluated on some other non-GPU/TPU hardware without massive parallelism”:**
>
> This is a great question. We believe that the throughput benefits of GDWS should appear regardless of the target hardware. The reason is that GDWS convolutions’ complexity is dominated by PW convolutions, which are very hardware friendly as compared to standard 2D convolutions even when their size and the number of MACs are similar. This is due to the fact that PW convolutions require minimal input data buffering. Each input pixel is used once for every filter as opposed to standard convolutions where the same pixel would be needed roughly $K^2$ times for a $K\times K$ convolution operation. This simplified mapping allows the hardware to carry the expensive PW layers without the overhead of input buffering.
>
> Note that we picked an NVIDIA Jetson as it is a very popular off-the-shelf hardware solution for AI on embedded platforms and requires minimal mapping effort.
>
> **“Furthermore, I have the feeling that the comparison with unstructured pruning is unfair. Unstructured pruning can benefit a lot if suitable sparse tensor structures are used.”:**
>
> Our work targets general hardware architectures. A limitation of unstructured pruning is that it requires custom hardware to fully leverage its sparse tensor structure. The benefit of GDWS is that it does not require any custom hardware/software support, which makes its deployment in the real world all the more easier. Note that one can also design custom hardware tailored for the GDWS structures as well, which can also boost performance theoretically.
>
> **“I believe that a network with 95% sparsity can have way more throughput than one without any sparsity...”:**
>
> Indeed it is surprising that a 95% pruned network admits no improvement in FPS when mapped onto hardware. We were initially surprised as well and conducted a detailed study shown in Section 2.3 of the supplementary material. We find that sparsity on its own is insufficient to boost FPS on typical hardware. It is the filter sparsity that needs to be high for FPS improvements to occur. In Figure 2 of Section 2.3 in the supplementary section, we plot the per-layer filter sparsity of HYDRA pruned VGG-16 and WideResNet-28-4 (publicly available thanks to the authors) at different pruning ratios: 90%, 95%, and 99%. It turns out, for both networks, only at extreme pruning ratios such as 99% does the filter sparsity in both networks appear to be significant, which translates to some improvement in FPS. The case for p=95% results in some filter pruning in the early layers where the weight dimensionality is small, but goes down to 0 as the network goes deeper. Thus, in reality, for a pruned convolutional layer to have significant improvement in performance on off-the-shelf hardware, it requires that a significant amount of filters be completely zero (i.e., can be pruned out).
>
> **“I would have liked to see some experiments that show how the individual layers are affected by the compression … It would also be interesting to see how the accuracy behaves for different g_c / different approximation errors”:**
>
> This is an excellent suggestion. We tabulate the per-layer average number of DW filters per channel ($g_c$) for VGG-16+GDWS on CIFAR-10 below. For each layer, the average $g_c$ is computed as $G/C$ where $G=\sum g_c$ and $C$ is the number of input channels. Keep in mind that $g_c <= K^2=9$. We also show these statistics for three different approximation errors $\beta$.
>
> |    Layer Index   |  L0  |  L1  |  L2  |  L3  |  L4  |  L5  |  L6  |  L7  |  L8  |  L9  |  L10 |  L11 |
> |:----------------:|:----:|:----:|:----:|:----:|:----:|:----:|:----:|:----:|:----:|:----:|:----:|:----:|
> | $\beta = 0.25$ | 6.44 | 7.78 | 7.97 | 8.52 | 8.54 | 8.58 | 8.40 | 5.07 | 1.90 | 2.29 | 2.79 | 5.60 |
> |  $\beta = 0.5$ | 5.88 | 7.33 | 7.56 | 8.30 | 8.32 | 8.37 | 8.08 | 4.12 | 1.30 | 1.60 | 2.28 | 5.11 |
> |   $\beta = 2$  | 4.36 | 5.94 | 6.23 | 7.53 | 7.55 | 7.59 | 6.88 | 1.80 | 1.30 | 1.30 | 1.38 | 4.03 |
>
> We also provide the associated FPS and $A_{\text{rob}}$ below:
>
> |           Model           | $A_{\text{rob}}$ | FPS |
> |:-------------------------:|:----------------------------:|:---:|
> |           VGG-16          |             49.05            |  36 |
> | + GDWS ($\beta = 0.25$) |             49.54            | 129 |
> |  + GDWS ($\beta = 0.5$) |             49.45            | 131 |
> |   + GDWS ($\beta = 2$)  |             45.41            | 140 |
>
> As expected, the average $g_c$ drops as the approximation error is allowed to increase (higher $\beta$). Furthermore, the early layers tend to have higher average $g_c$ which is consistent with the well-known phenomenon of errors getting amplified as they propagate through the network. Note that we can infer the per-layer complexity reduction by simply looking at $G/C$ of each layer (please refer to eq (3) of the paper). We will incorporate these results in our paper (or supplementary material if space is an issue).
>
> **"From the reported FPS it is difficult to tell where the increased throughput stems from. Maybe you can also provide some intuition why the FPS is affected more severely than the size.":**
>
> FPS is affected more than model size because GDWS convolutions are much more hardware friendly than standard 2D convolutions even at iso complexity (model size or the number of operations). The reason for this is that PW convolutions, which dominate the complexity in GDWS, require minimal input data buffering as mentioned before.
>
> **“Is the "natural accuracy" the accuracy on the standard test dataset (without adversarial perturbations)? In this case, can you comment on the rather weak natural accuracies on the given datasets”:**
>
> Yes, natural accuracy (or $A_{\text{nat}}$) is the accuracy on the standard test set, i.e., without any adversarial perturbations. The reason the $A_{\text{nat}}$ numbers seem weak, for example 82.41% for ResNet-18 on CIFAR-10, is because these models are adversarially trained and not vanilla trained. If one were to do vanilla training, i.e., train the models on standard unperturbed training images, then one would expect $A_{\text{nat}}$ of >90% for CIFAR-10, but its robust accuracy would be close to 0%. It is well known [1][2][3] that adversarially trained models, while significantly more robust to adversarial perturbations than vanilla trained ones, suffer a drop in A_nat as a consequence.
>
> [1] Zhang, Hongyang, et al. "Theoretically principled trade-off between robustness and accuracy." International Conference on Machine Learning. PMLR, 2019.
>
> [2] Tsipras, Dimitris, et al. "Robustness May Be at Odds with Accuracy." International Conference on Learning Representations. No. 2019. 2019.
>
> [3] Madry, Aleksander, et al. "Towards Deep Learning Models Resistant to Adversarial Attacks." International Conference on Learning Representations. 2018.
>
> **“Figure 1a: What are the numbers in brackets?”:**
>
> The numbers in brackets are the compression ratios of each method with respect to the AT baseline (black diamond). This is stated in the last sentence in the caption of Fig.1.

---

> > ### Comment · Reviewer_c4Yo · 2021-08-30
> > **Response**
> >
> > I thank the authors for their rebuttal. Most of my concerns have been addressed. I still think that training with GDWS from scratch would be very interesting and I think some very simple choices for g (e.g., uniform across all layers) would be interesting first experiments. However, I also agree that this would be a perfect opportunity to continue this line of research. I still do not fully understand why some common operations facilitating sparse tensor operations if operated on a conventional CPU would not provide any benefit (for sparsity ~90%). Please state in the next version of the paper more explicitly that the adversarial robustness of GDWS is due to the approximation quality and not to any other particular properties of GDWS. I think the paper provides valuable contributions and remain with my score in favor of acceptance.

---

> > > ### Author Response · Authors · 2021-08-30
> > > **Response to Reviewer c4Yo**
> > >
> > > We thank you for your careful reading of our paper and your strong support of our work. In the final version of the paper, we will most certainly state that the adversarial robustness benefits of GDWS are due to its approximation quality and incorporate other clarifications arising from this review process.

---

### Official Review · Reviewer_n5ng · 2021-07-16

**Rating:** 6
**Confidence:** 3

**Summary:**

This work proposes a scalable and universal post-training approximation of standard 2D convolutions, namely Generalized Depthwise-Separable (GDWS) convolutions, to improve the throughput (FPS) of adversarial robust neural networks (NNs) on real-life hardware. Experiments are performed on different datasets, including CIFAR-10, SVHN, and ImageNet, with different network architectures, including VGG-16 and several ResNet variants.

**Limitations And Societal Impact:**

There are some limitations:

1. In Table 3, GDWS uses a fixed setting for each architecture. With VGG-16, the FPS of GDWS is outperformed by ADMM with $p=75\%$. Although GDWS has better accuracy than ADMM ($p=75\%$), I still wondering if the FPS is further improved, whether GDWS can still outperform ADMM.

2. In Table 4, when GDWS is applied on a highly compressed network, the parameter size is increased, which is an undesired behavior.

A minor issue: Table 5 should be with section 4.2 but is misplaced in section 4.3.

**Main Review:**

The GDWS convolution improves the throughout of adversarial robust NNs, which is always ignored by existing pruning-based compression methods. The method directly operates on pre-trained models without additional training, which makes it more efficient and easier to scale up than pruning methods. Experiments are performed on four different network architectures and is also compared to a NAS method and several structured and unstructured pruning methods.

**Time Spent Reviewing:**

5

---

> ### Author Response · Authors · 2021-08-10
> **Reply to Reviewer n5ng**
>
> Thank you for the positive review. We hope that we address your comments below:
>
> **“Although GDWS has better accuracy than ADMM (p=75), I still wondering if the FPS is further improved, whether GDWS can still outperform ADMM.”:**
>
> That is a very good question. One would expect that GDWS’ FPS could be improved by allowing for higher approximation error via Algorithm 3. In fact, we study this effect in Figure 1b & 1c of the supplementary material where we plot $A_{\text{rob}}$ and $A_{\text{nat}}$ vs FPS for a pre-trained VGG-16 on CIFAR-10. One can see that FPS does improve to some extent but after a certain point there is a rapid drop in robustness. We ran additional experiments to compare with ADMM using the same VGG-16 baseline from Table 3 and find similar conclusions. Bottomline is that ADMM benefits from training in the high FPS (> 130) regime. It will be interesting to see in a future work if training with GDWS can achieve superior performance in this regime:
>
> |           Model           | $A_{\text{rob}}$ | FPS |
> |:-------------------------:|:----------------------------:|:---:|
> |           VGG-16          |             45.78            |  36 |
> |  + GDWS ($\beta = 0.1$) |             45.97            | 104 |
> |  + GDWS ($\beta = 0.5$) |             46.28            | 119 |
> | + GDWS ($\beta = 1.25$) |             44.62            | 129 |
> |  + GDWS ($\beta = 2.5$) |             36.06            | 135 |
>
> **“In Table 4, when GDWS is applied on a highly compressed network, the parameter size is increased, which is an undesired behavior.”:**
>
> Note our baseline recommendation is to apply GDWS to pre-trained unpruned models in order to preserve their robustness, improve FPS, and decrease the model size (as seen in Table 1). Application of GDWS to HYDRA was done just for completeness because their pruned models showed little to no improvement in FPS (as seen in Table 4) while incurring massive drops in robustness. Indeed, when GDWS is applied to HYDRA pruned models we do observe that the resultant models achieve a very good trade-off between FPS and model size, at iso-robustness.
>
> **“A minor issue: Table 5 should be with section 4.2 but is misplaced in section 4.3”:**
>
> Thank you for pointing this out. We will adjust the section of Table 5 accordingly.

---

### Official Review · Reviewer_Sfha · 2021-07-17

**Rating:** 7
**Confidence:** 4

**Summary:**

This paper proposes the method of Generalized Depthwise-Separable (GDWS) convolution for robust model compression. It tries to improve frames-per-second on real-life hardware while simultaneously preserving robustness to adversarial perturbations.

**Limitations And Societal Impact:**

I don't see any potential negative societal impact of their work.

**Main Review:**

Strength:
+ The paper is clearly written with sound theoretical proof.
+ Good design of experiments with different models for comparison.

Weaknesses:
- The results look good in general, but for Table 2, RobNet results look weird. As in the original RobNet paper, it is much stronger than Resnet in terms of perturbations. But in Table 2, it seems that RobNet underperforms ResNet baseline. I am wondering if the same backbone are used for RobNet? Is it a fair comparison?
- Only PGD are present in the experiments. I am wondering how the proposed method performs under different types of attacks. Then the robustness of the proposed method would be better evaluated.



**Time Spent Reviewing:**

2 hours

---

> ### Author Response · Authors · 2021-08-10
> **Reply to Reviewer Sfha**
>
> Thank you so much for the positive review. We hope that we address your comments below:
>
> **“The results look good in general, but for Table 2, RobNet results look weird. As in the original RobNet paper, it is much stronger than Resnet in terms of perturbations. But in Table 2, it seems that RobNet underperforms ResNet baseline. I am wondering if the same backbone are used for RobNet? Is it a fair comparison?”:**
>
> Yes, in our comparison with RobNet [7] in Table 2 of our paper, we chose the best performing RobNet denoted by RobNet-free in Table 1 of [7]. The authors were kind enough to release that model on GitHub, so we did not train any RobNets on our end.
>
> The reason RobNet seems to underperform ResNet in Table 2 is because we had to train the ResNet models ourselves before applying GDWS since the authors of [7] did not release their ResNet models. We chose two ResNet variants: ResNet-50 (the pre-activation version) and a standard WideResNet-28-4, both of which the original RobNet authors did not use in their comparison. In order to make the comparison fair, we also trained our models with adversarial training via PGD-7 (7 iterations) and the same hyperparameters (learning rate, number of epochs, optimizer, …).
>
> **“I am wondering how the proposed method performs under different types of attacks”:**
>
> This is a great suggestion. We used PGD-generated adversaries only in this paper as it has been well known to be one of the most effective adversaries. Nevertheless, we conduct an extra set of attacks, highlighted in the Table below, on the VGG-16 network on CIFAR-10 (baseline from Table 1 of our paper). We use the foolbox (https://github.com/bethgelab/foolbox) implementation of all these attacks to ensure proper implementation. All the attacks are using $\ell_\infty$-bounded perturbations with $\epsilon=8/255$, similar to our PGD results  in the paper. As expected, GDWS preserves the robustness of the pre-trained baseline, across different attack methods. We will include these results in our paper or supplementary material if space is an issue.
>
>
> |          Model          |  FGSM |  BIM  | DeepFool | FPS |
> |:-----------------------:|:-----:|:-----:|:--------:|:---:|
> |          VGG-16         | 52.53 | 49.61 |   47.89  |  36 |
> | + GDWS ($\beta=0.25$) | 53.19 | 50.08 |   47.28  | 129 |
> |  + GDWS ($\beta=0.5$) | 52.69 | 49.87 |   46.32  | 131 |

---

### Decision · Program_Chairs · 2021-09-28

**Decision:**

Accept (Spotlight)

**Comment:**

Taken at face value, this paper seems to have a lot of variance in the scores, but this is driven mostly by the low score given by reviewer PpQM. The authors have noted that PpQM's review does not make sense, I have read it and confirmed that it does not make sense, and in fact another reviewer has confirmed to me that it does not make sense. That makes it especially strange that they have chosen to respond to the rebuttal that their score is unchanged, but at any rate I think it's safe to ignore their review.

With that out of the way, the scores are 6, 6, and 7.
I found the review of c4Yo (a 6) most informative, and appreciated the discussion that followed.
I will recommend acceptance in this instance

**Consistency Experiment:**

NeurIPS has a long history of experimentation. In 2014, NeurIPS ran an experiment in which 10% of submissions were reviewed by two independent committees to quantify the randomness in the review process. This year, we repeated a variant of this experiment to see how the quality of the review process has changed over time.  This paper was part of the experiment and was therefore assigned to two committees (consisting of reviewers, an Area Chair, and a Senior Area Chair) that reached independent decisions.  If both committees made the same recommendation, this recommendation was followed. If a single committee recommended acceptance, the paper was accepted (with the exception of a few cases in which the other committee identified what we considered a fatal flaw, e.g., an error in a key result).

This copy’s committee reached the following decision: **Accept (Poster)**

The other committee assigned to the paper recommended **Accept (Spotlight)**.  You can find the other set of reviews, along with any follow up discussion with the authors here:
https://openreview.net/forum?id=jNq-i1zd0t9